# Improving the oxygen redox reversibility of Li-rich battery cathode materials via Coulombic repulsive interactions strategy

Qingyuan Li [1], De Ning[2,3], Deniz Wong[2], Ke An [4], Yuxin Tang[5], Dong Zhou[2], Götz Schuck[2], Zhenhua Chen[6], Nian Zhang [7] & Xiangfeng Liu [1,8 ✉]

The oxygen redox reaction in lithium-rich layered oxide battery cathode materials generates extra capacity at high cell voltages (i.e., >4.5 V). However, the irreversible oxygen release causes transition metal (TM) dissolution, migration and cell voltage decay. To circumvent these issues, we introduce a strategy for tuning the Coulombic interactions in a model Li-rich positive electrode active material, i.e., $Li_{1.2}Mn_{0.6}Ni_{0.2}O_2$. In particular, we tune the Coulombic repulsive interactions to obtain an adaptable crystal structure that enables the reversible distortion of $TMO_6$ octahedron and mitigates TM dissolution and migration. Moreover, this strategy hinders the irreversible release of oxygen and other parasitic reactions (e.g., electrolyte decomposition) commonly occurring at high voltages. When tested in non-aqueous coin cell configuration, the modified Li-rich cathode material, combined with a Li metal anode, enables a stable cell discharge capacity of about 240 mAh g$^{-1}$ for 120 cycles at 50 mA g$^{-1}$ and a slower voltage decay compared to the unmodified $Li_{1.2}Mn_{0.6}Ni_{0.2}O_2$.

[1] Centre of Materials Science and Optoelectronics Engineering, College of Materials Science and Optoelectronic Technology, University of Chinese Academy of Sciences, Beijing 100049, P. R. China. [2] Department of Dynamics and Transport in Quantum Materials and Department of Structure and Dynamics of Energy Materials, Helmholtz-Zentrum Berlin für Materialien und Energie, Hahn-Meitner-Platz 1, Berlin 14109, Germany. [3] Centre for Photonics Information and Energy Materials, Shenzhen Institutes of Advanced Technology, Chinese Academy of Sciences, Shenzhen 518055, P.R. China. [4] Neutron Scattering Division, Oak Ridge National Laboratory, Oak Ridge, TN 37830, USA. [5] College of Chemical Engineering, Fuzhou University, Fuzhou 350116, P. R. China. [6] Shanghai Synchrotron Radiation Facility, Shanghai Institute of Applied Physics, Chinese Academy of Sciences, Shanghai 201204, P. R. China. [7] Shanghai Institute of Microsystem and Information Technology, Chinese Academy of Sciences, Shanghai 200050, P. R. China. [8] CAS Centre for Excellence in Topological Quantum Computation, University of Chinese Academy of Sciences, Beijing 100190, China. ✉email: liuxf@ucas.ac.cn

Lithium-ion batteries (LIBs) play a vital role as rechargeable power sources in various electronic devices and facilities today; in particular, they are used in electric vehicles (EVs) owing to their high specific energy, high potential, and light weight characteristics[1–3]. In LIBs, cathode materials largely determine the specific energy, lifespan, and safety because they have a lower specific capacity and unstable structure compared with anode materials. Therefore, the development of high-performance cathode materials is extremely urgent. In all kinds of cathode materials, Li-rich Mn-based oxides are one of the most promising candidates for next-generation cathode materials due to their high specific capacity (>300 mAh g$^{-1}$) and high specific energy (~1000 Wh kg$^{-1}$), which originate from the charge compensation of both the anion and cation redox chemistry in Li-rich Mn-based oxides. However, some critical issues, such as structural instability, poor rate capability, and voltage decay, need to be resolved. In particular, continuous voltage decay and irreversible oxygen release are the main stumbling blocks for commercialization since they gradually reduce the energy content and lifetime during cycling. Safety concerns caused by oxygen evolution are also critical for Li-rich Mn-based oxides. To overcome these issues, numerous research efforts have been conducted to unravel the reasons for voltage fade and irreversible anion redox chemistry, and search for modification strategies.

Currently, it is well accepted that the layered to spinel phase transition caused by transition metal (TM) migration is the primary origin for voltage decay[4–6]. It is worth noting that this process is closely related to the oxygen release induced by the irreversible oxygen redox chemistry because the structural instability and oxygen release from oxygen redox chemistry can accelerate voltage fading[7–10]. Therefore, the study and modulation of anion redox chemistry is urgent. To better modulate the anion redox activity and reversibility, scientists have proposed covalence theory[11,12] and unhybridized oxygen 2p bands[13]. In particular, the latest O nonbonding states theory has been widely accepted because it can effectively rationalize anion redox chemistry[12,13]. Recent studies report that the electrochemical property of this O nonbonding state is related to the relative positions of the antibonding (TM-O)* and O 2p nonbonding bands. However, the band position is determined by the introduction of the $d$–$d$ Coulomb interaction term $U$ and charge transfer (CT) term $\Delta$ from solid-state physics[12,14,15]. According to this theory, electrons are extracted from the filled lower-Hubbard bands (LHB) caused by Mott-Hubbard splitting in many oxides and fluorides due to $U << \Delta$ (Fig. 1a), which is known as cationic redox chemistry. In contrast, electrons are removed from the unhybridized O 2p state located above the LHB and are accompanied by serious oxygen release when $U >> \Delta$ for most Li-rich cathode materials, which is called irreversible oxygen redox (Fig. 1b). For the middle case, $U \approx 2\Delta$, electrons can exchange from both the nonbonding O 2p state and (TM-O)* band due to the overlap between the LHB and unhybridized O 2p bands (Fig. 1c). Therefore, the capacity can be increased from both the anionic and cationic redox reactions while mitigating oxygen release, meaning that oxygen redox is more reversible[12,15]. Hence, to obtain reversible anion redox and decrease structural damage, strategies to regulate $U$ and $\Delta$ are urgently needed. One is to choose a TM with a higher $d$ state, such as Ru and Ir, to reduce the $U$ term; as a result, the LHB can overlap with the nonbonding O 2p band better, and the anion redox chemistry will be enhanced while reducing O$_2$ escape. The other is to substitute O atoms with less electronegative elements, such as S and Se, which can make ligands with nonbonding p states approach the TM band. However, neither strategy is suitable for practical application due to their high cost or low voltage plateau. It is worth noting that Li$_{1.2}$Mn$_{0.6}$Ni$_{0.2}$O$_2$ with oxygen vacancies and/or a spinel phase

can improve the properties of cathode materials. Nonetheless, until now, to the best of our knowledge, the study of mechanisms is still insufficient, especially from the direction of the band position combined with experiments.

Herein, we reveal that the enhancement of oxygen redox reversibility and the mitigation of TM migration in Li-rich Mn-based oxide (Li$_{1.2}$Mn$_{0.6}$Ni$_{0.2}$O$_2$) materials with oxygen vacancies are closely related to the adjustment of $d$–$d$ Coulomb interaction $U$. The enhanced $d$–$d$ excitation of Mn and the lowering of term $U$ can be verified by means of resonance inelastic X-ray scattering (RIXS) with Li$_{1.2}$Mn$_{0.6}$Ni$_{0.2}$O$_2$ due to the reduction of Mn and the incorporation of oxygen vacancies, which illustrates the strength of reversible oxygen redox while reducing O$_2$ release. Both of the above results are validated by electrochemical energy storage characterization and differential electrochemical mass spectrometry (DEMS) measurements. We also report that the softened crystal structure is favorable to the reversible distortion of TMO$_6$ octahedra in TM layers rather than the migration of TMs caused by a broken rigid octahedral structure; therefore, the voltage fade can be mitigated. Furthermore, based on the experimental results, the general enhancement mechanism for the reversible anion redox chemistry is rationalized by the schematic density of state (DOS) patterns.

## Results

**Creation of oxygen vacancies and reduced manganese.** To study the effect of oxygen vacancies and spinel-like structures in the pristine Li$_{1.2}$Mn$_{0.6}$Ni$_{0.2}$O$_2$ (LRMO) cathode, a modified sample Li$_{1.2}$Mn$_{0.6}$Ni$_{0.2}$O$_{2-\delta}$ (M-LRO) with oxygen vacancies and reduced Mn is synthesized by a more uniform liquid–solid method. First, we used X-ray powder diffraction (XRD) to determine the crystal structure of pristine and Li$_{1.2}$Mn$_{0.6}$Ni$_{0.2}$O$_{2-\delta}$, and the refined patterns of both samples are shown in Supplementary Fig. 1. The results illustrate that both samples are composed of layered rhombohedral phases with the space group of $R\bar{3}m$, but peaks of the spinel phase are not found. There are some small superlattice peaks at approximately 20°, which are caused by the monoclinic phase ($C2/m$); however, these are not refined because of their low content. Moreover, neutron powder diffraction (NPD) is used to quantitatively detect the amount of oxygen vacancies due to its sensitivity to light elements. Figure 1d, e show the refined patterns, and f and g show the crystal models and corresponding lattice parameters. As a result, oxygen vacancies account for approximately 5% of the Li$_{1.2}$Mn$_{0.6}$Ni$_{0.2}$O$_{2-\delta}$ sample. Based on the refined results, we simulated the effect of different oxygen vacancy concentrations on the structure, as shown in Supplementary Fig. 1d. By comparing with Supplementary Fig. 1c, we find that 5% oxygen vacancies have almost no significant effect on the crystal structure. Furthermore, the lattice parameters and atomic occupation information of both samples are shown in Supplementary Table 1 and Supplementary Table 2, respectively. We can find that the lattice parameters and TM–O bond length are increased for Li$_{1.2}$Mn$_{0.6}$Ni$_{0.2}$O$_{2-\delta}$ due to the reduction expansion[16,17], which indicates the existence of crystal distortion. Moreover, the increased Li$^+$ diffusion channel is beneficial to a high rate capability. Spherical aberration-corrected scanning transmission electron microscopy (SAc-STEM) was used to observe the atomic arrangement. As Supplementary Fig. 2 shows, the high-angle annular dark-field scanning transmission electron microscopy (HAADF-STEM) images indicate that the bulk structure is not destroyed by hydrazine aqueous solution, but some zigzag atomic ladders are produced on the surface of the Li$_{1.2}$Mn$_{0.6}$Ni$_{0.2}$O$_{2-\delta}$ sample due to the oxygen vacancies, which can provide more active sites and reduce the thickness of the cathode electrolyte interphase (CEI) during extended cycling.

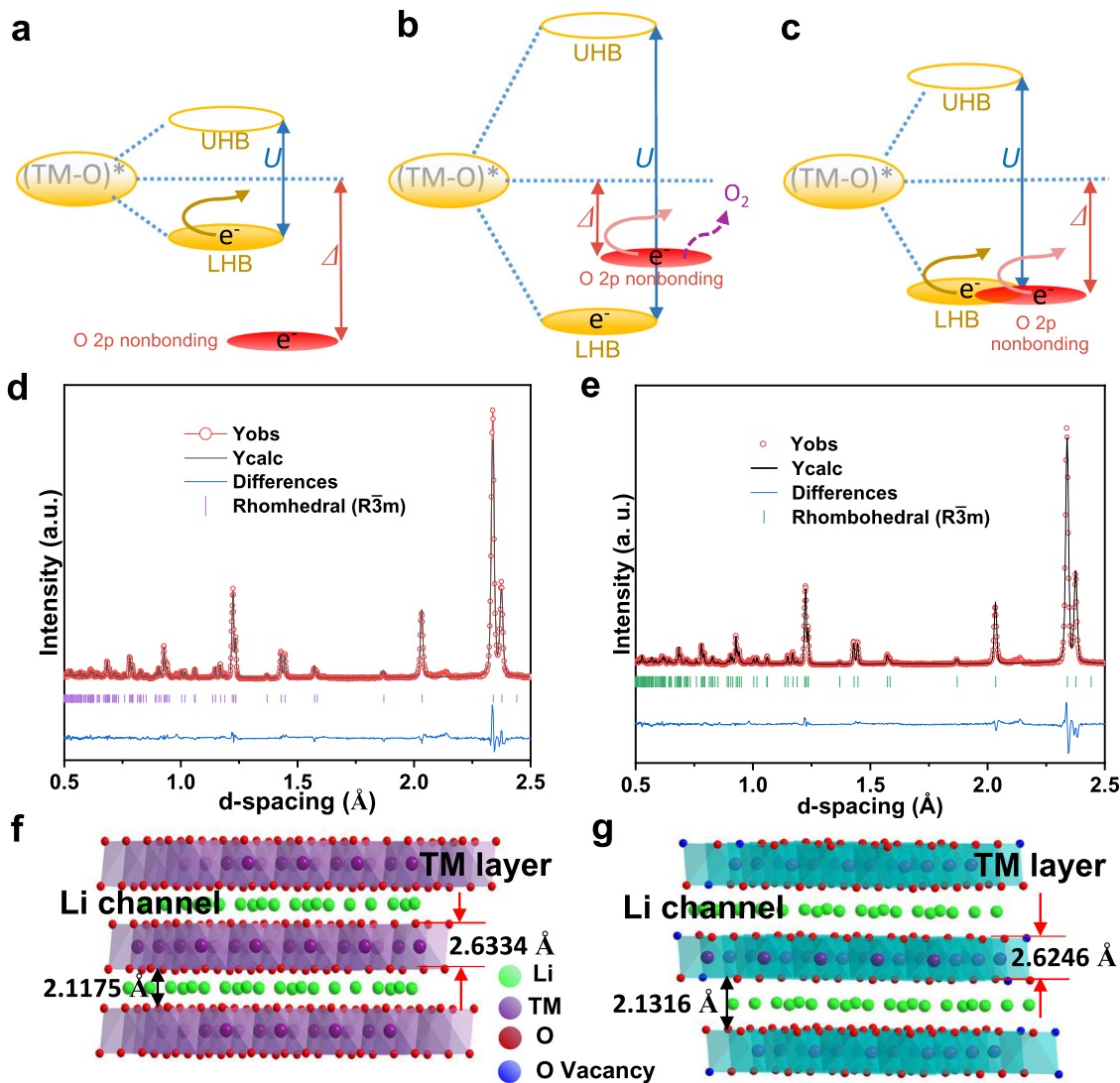

**Fig. 1 Anion redox mechanism and neutron powder diffraction refinement patterns with corresponding crystal models. a–c** Qualitative analysis of the relationship between the *d–d* Coulombic interaction term *U* and the charge transfer term *Δ*. Electrons are removed from the filled LHB band when *Δ* >> *U* (**a**). Electrons leave the O 2*p* nonbonding band accompanied by $O_2$ release, which indicates irreversible anion redox chemistry when *U* >> *Δ* (**b**). For the middle case, *U* ≈ 2*Δ*, electrons can be removed from both bands and prevent $O_2$ release when $Li^+$ is extracted (**c**). TOF NPD refinement patterns of pristine $Li_{1.2}Mn_{0.6}Ni_{0.2}O_2$ (**d**) and $Li_{1.2}Mn_{0.6}Ni_{0.2}O_{2-\delta}$ (M-LRO) samples (**e**). According to the refinement results, the oxygen vacancy is approximately 5% of the M-LRO sample, and the corresponding refined data are marked in the crystal models: **f** pristine and **g** M-LRO. The term a.u. means arbitrary units.

Moreover, there are relatively more dislocations in the $Li_{1.2}Mn_{0.6}Ni_{0.2}O_{2-\delta}$ sample (Supplementary Fig. 2c, d); therefore, although this requires further study, we think that these boundary defects and dislocations are the causes of crystal structure softening[18,19]. The [7]Li magic angle spinning (MAS) solid-state nuclear magnetic resonance (ssNMR) measurements are shown in Supplementary Fig. 3, where the centre peaks at 0 ppm are caused by $Li_2CO_3$, LiOH and $Li_2O$ on the surface and located in a diamagnetic environment[20]. The oxygen vacancy could be illustrated by the widened sideband pattern in the $Li_{1.2}Mn_{0.6}Ni_{0.2}O_{2-\delta}$ sample. Because oxygen vacancies affect the asymmetric distribution of the electron cloud around the nucleus, the local magnetic field around the nucleus is perturbed, thereby affecting the anisotropy of the chemical shift of the nucleus, which will broaden the spectrum peak[21,22]. In addition to the production of oxygen vacancies, tetravalent manganese can also be reduced, which can be confirmed by the calculation results of the fitted Mn 3*s* spectra, obtained by X-ray photoelectron spectroscopy (XPS)[23] and Mn *L*-edge spectra of soft X-ray absorption spectroscopy (XAS, Supplementary Fig. 4). According to the

formula between the Mn valence ($v_{Mn}$) and splitting energy of Mn 3*s*[24], $v_{Mn} = 9.67-1.27\Delta E_{3s}$/eV; thus, the Mn valence in the $Li_{1.2}Mn_{0.6}Ni_{0.2}O_{2-\delta}$ sample is +3.83. This result is perfectly in line with the oxygen vacancy (5%) determined by neutron diffraction (Supplementary Table 2). Moreover, the dashed line shows the existence of $Mn^{3+}$ on the surface of the $Li_{1.2}Mn_{0.6}Ni_{0.2}O_{2-\delta}$ sample by soft XAS in total electron yield (TEY) mode (Supplementary Fig. 4c). It is worth noting that the bulk structure is uninfluenced for both samples, as shown by the partial fluorescence yield (PFY) mode because of its detection depth of 100 nm.

**Electrochemical characterizations**. Based on the materials and analysis above, the electrochemical characterizations of both samples were performed. First, we tested the rate capacity of different samples treated with different concentrations of aqueous hydrazine for different times. We find that the sample obtained after treatment with a 2 M hydrazine aqueous solution for 1 h exhibits the best rate capacity (Supplementary Fig. 5). Hence, we only focus on this sample and the pristine material in the

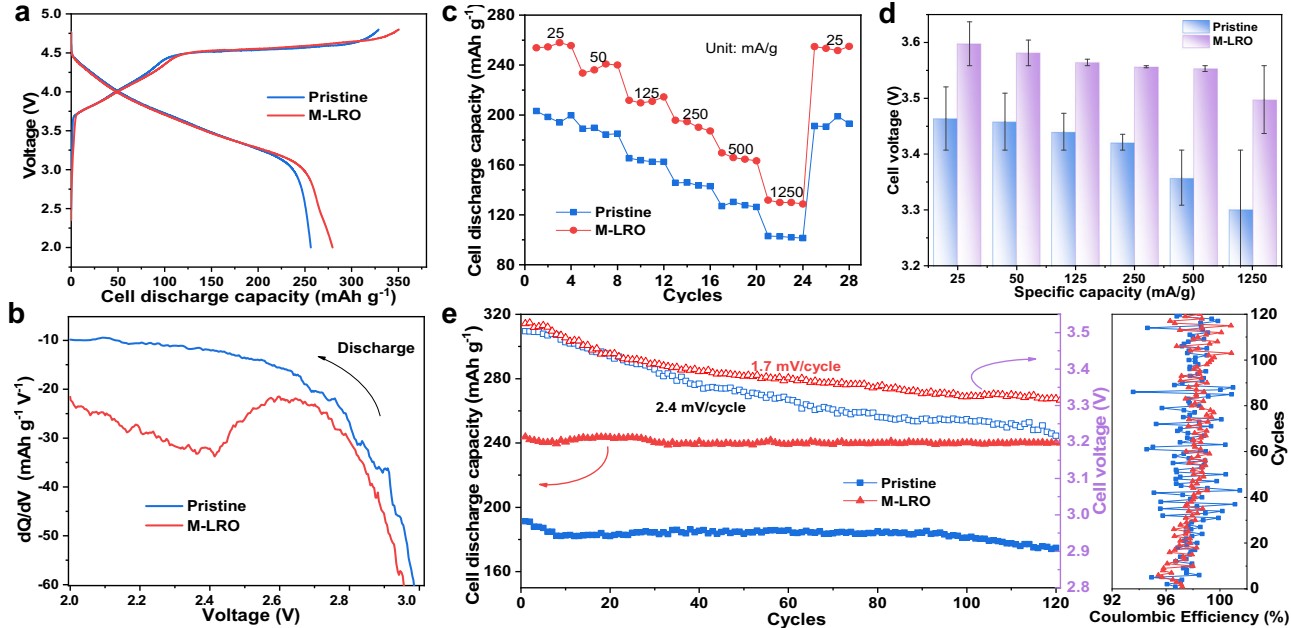

**Fig. 2 Electrochemical energy storage performance of Li-rich Mn oxide-based layered cathode materials.** Initial charge–discharge curves of the pristine and M-LRO samples at a specific current of 12.5 mA/g. There is a slight plateau at 2.4 V for the M-LRO sample due to the existence of reduced Mn (**a**), which can be verified by the dQ/dV curves (**b**). Discharge capacity of both samples at various specific currents (**c**). Median discharge voltage of both samples at different rates (**d**). Cycling performance and corresponding voltage decay at a specific current of 50 mA/g for the pristine $Li_{1.2}Mn_{0.6}Ni_{0.2}O_2$ and $Li_{1.2}Mn_{0.6}Ni_{0.2}O_{2-\delta}$ (M-LRO) samples. The right panel is the corresponding Coulombic efficiency (**e**).

following tests to study the mechanisms. There is a slightly visible plateau at ~2.4 V for the modified sample in the initial discharge curves and the dQ/dV profile in Fig. 2a, b, which is related to the reduction or activation of $Mn^{4+/3+}$ and confirms the existence of $Mn^{3+}$ in the initial state[25,26]. Figure 2c shows the rate capacity of both samples. The specific capacity contribution is more than 260 mAh/g at a specific current of 25 mA/g for the $Li_{1.2}Mn_{0.6}Ni_{0.2}O_{2-\delta}$ sample, which is ascribed to the enhanced anion and cation redox chemistry. In comparison, the discharge capacity of the pristine sample is only ~200 mAh/g at the same specific current. The leading discharge capacity is still maintained at a high specific current for the modified material. In addition to the excellent capacity in the $Li_{1.2}Mn_{0.6}Ni_{0.2}O_{2-\delta}$ sample, the voltage decay is significantly mitigated at different specific currents for $Li_{1.2}Mn_{0.6}Ni_{0.2}O_{2-\delta}$ (Fig. 2d). Specifically, the median discharge voltage of the pristine sample is lower than that of the $Li_{1.2}Mn_{0.6}Ni_{0.2}O_{2-\delta}$ sample at any specific current. The electrochemical performance is also illustrated by the cycling stability at different specific currents. Figure 2e shows the cycling property, voltage decay behavior and Coulombic efficiency of both samples at a specific current of 50 mA/g. A negligible capacity fade can be observed during 120 cycles for the $Li_{1.2}Mn_{0.6}Ni_{0.2}O_{2-\delta}$ material. The modified $Li_{1.2}Mn_{0.6}Ni_{0.2}O_{2-\delta}$ sample has less voltage decay with an attenuation value of only 1.7 mV per cycle for the, than the pristine sample (2.4 mV per cycle). This improvement in electrochemical performance can also be reflected by the smaller fluctuation in the Coulombic efficiency of the modified sample in the right panel of Fig. 2e. In addition to the cycling performance at a low specific current, the cycling performance at specific currents of 250 mA/g and 1250 mA/g are investigated (Supplementary Fig. 6a, b). As the results show, the capacity of $Li_{1.2}Mn_{0.6}Ni_{0.2}O_{2-\delta}$ is higher and more stable than that of the pristine sample. For the $Li_{1.2}Mn_{0.6}Ni_{0.2}O_{2-\delta}$ sample, the voltage decay is only 1.0 and 0.54 mV/cycle at specific currents of 250 mA/g and 1250 mA/g, respectively. In contrast, the voltage attenuation of the pristine sample is more serious, and the decay

value reaches 1.8 and 0.68 mV/cycle at the same specific current. Additionally, the Coulombic efficiency patterns at different specific currents are shown in Supplementary Fig. 6c, d. Clearly, the modified sample has better Coulombic efficiency with an average greater than 99% after long-term cycling. The above rate performance is in good agreement with the result of the potentiostatic intermittent titration technique (PITT) in Supplementary Fig. 7 because the large lithium-ion diffusion coefficient allows the battery to undergo fast charging; the detailed discussion is shown in Supplementary Fig. 7.

**Reversible distortion driven by reduced Mn and O vacancies.** To reveal the mechanisms of the mitigated voltage decay and increased capacity, in situ XRD measurements are carried out to monitor the crystal structure change in real time. The overall contour patterns of the pristine and $Li_{1.2}Mn_{0.6}Ni_{0.2}O_{2-\delta}$ samples are presented on the left of Fig. 3a, b. The asterisk indicates the Be/BeO peaks from the window of the cell mould. The representative diffraction peaks are indexed with (003), (101) and (104). The change in amplitude of these three peaks is smaller for the $Li_{1.2}Mn_{0.6}Ni_{0.2}O_{2-\delta}$ sample than for the pristine sample. To clearly observe the change, they are extracted from the overall pattern and inspected, as shown in the middle of Fig. 3a, b. For both samples, the (003) peak shifts to a lower 2θ degree; however, the (101) and (104) peaks shift to a higher degree during the charging process, which is attributed to the increased lattice parameter *c* caused by the promoted electrostatic repulsion of adjacent oxygen layers and the shrinkage of the TM–TM distance caused by the shortened TM–O bond due to $Li^+$ extraction, respectively[27]. Moreover, the oxidation of low valence state TMs with larger radii will also cause the layer spacing to contract[28]. It is worth noting that the peaks are almost stable for the $Li_{1.2}Mn_{0.6}Ni_{0.2}O_{2-\delta}$ sample from 4.5 to 4.8 V, which indicates less oxygen release and more active reversible oxygen redox chemistry than the pristine sample. During discharge, the diffraction peaks

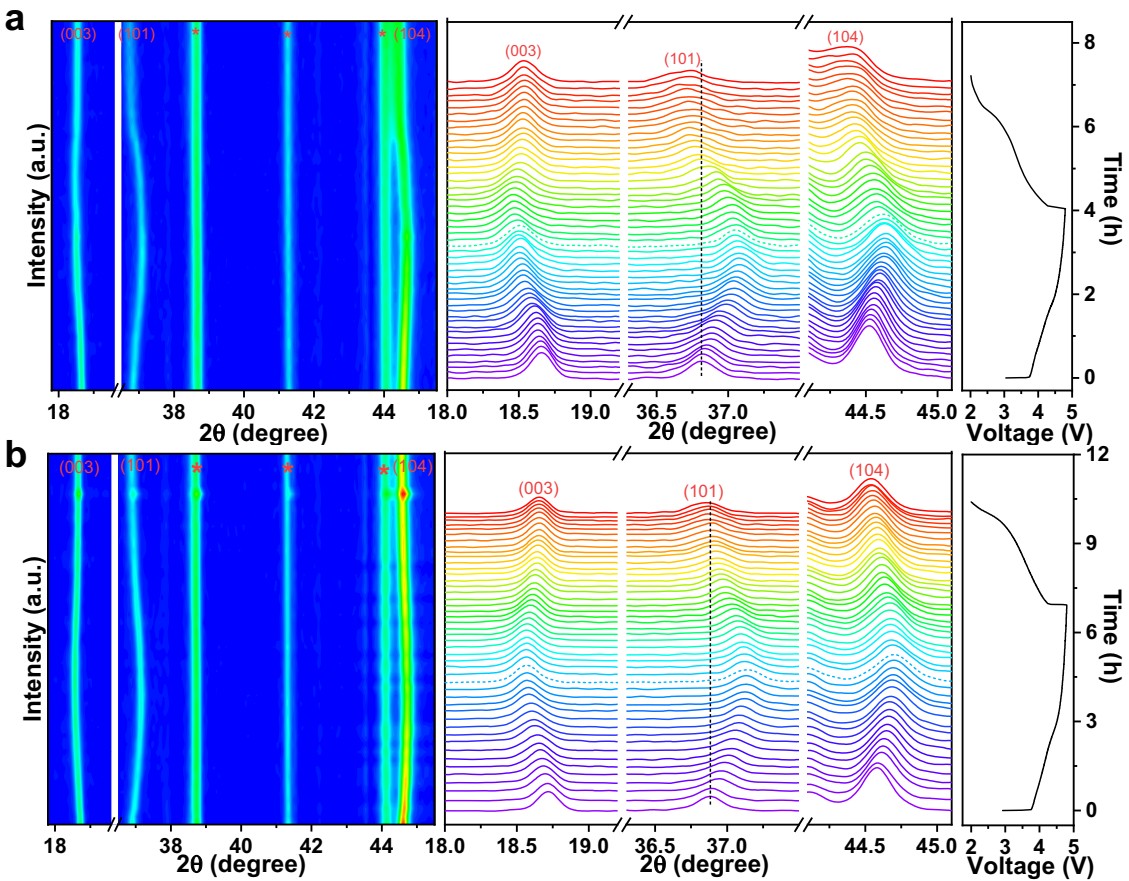

**Fig. 3 Crystal structure of Li-rich Mn oxide-based layered cathode materials.** In situ XRD of the pristine $Li_{1.2}Mn_{0.6}Ni_{0.2}O_2$ (**a**) and $Li_{1.2}Mn_{0.6}Ni_{0.2}O_{2-\delta}$ (M-LRO) samples (**b**) in the first cycle. The left patterns are 2D contours, and the asterisk label indicates the diffraction peaks of Be in the battery window. The vertical dotted lines illustrate the reversible and irreversible change of the (101) peak for the M-LRO and pristine samples. The right spectra show the initial charge–discharge curves. The term a.u. means arbitrary units.

of both samples shift to a reverse direction accompanied by $Li^+$ insertion. However, the peaks of the pristine sample cannot move to the original position, especially for the dashed line shown, which is caused by irreversible oxygen loss and irreversible TM migration from the octahedral sites of the rigid TM layers to the octahedral sites of the $Li^+$ layers, resulting in structural changes and voltage decay. In comparison, although it is thermo-dynamically favorable for TMs to migrate from TM layers to $Li^+$ layers, the softened crystal structure can bind TM migration in the form of a $TMO_6$ ($MnO_6/NiO_6$) octahedral distortion to prevent TMs from migrating to the $Li^+$ diffusion tunnels. This stabilizes the crystal structure and oxygen, which is confirmed by the Mn migration barrier in the following discussion. As a result, the voltage fade is mitigated.

In addition to the crystal structure, to illustrate the voltage decay mitigation by the flexible distortion of the softened structure due to oxygen vacancies, the stable electronic structure can be used to describe the reasons for voltage fade mitigation. The Mn $K$-edge spectra of both samples at different charge states are given in Fig. 4a, b. Accompanied by $Li^+$ deintercalation during charging, there is a weak continuous change in the Mn $K$-edge spectra of the pristine sample in Fig. 4a, which means that the electronic structure has been changed slightly but the valence state of manganese is stable. In contrast, the change in the absorption edges is more obvious for the $Li_{1.2}Mn_{0.6}Ni_{0.2}O_{2-\delta}$ sample in Fig. 4b, which indicates that the electronic structure and valence state of the Mn ions have both been changed when charged. The insets show the pre-edge absorption peaks, which

illustrate that the $MnO_6$ octahedral distortion is caused by the hybridization of the $3d-4p$ orbitals[29,30]. We find that the change is irreversible when the voltage is returned to 2.0 V from the charged 4.8 V in the pristine sample. We also quantitatively calculated the change in the pre-edge peaks at the charge–discharge states (Supplementary Fig. 8a). The intensities of both samples increase continuously with $Li^+$ deintercalation and reach a maximum when charged to 4.8 V. However, the change in amplitude of the pristine sample is more serious than that of the $Li_{1.2}Mn_{0.6}Ni_{0.2}O_{2-\delta}$ sample. In particular, the signal cannot return to the original position when the pristine sample is discharged to 2.0 V. Moreover, we define here that the intensity of structural distortion is proportional to the change in intensity of the pre-edge peaks; therefore, the corresponding extent of distortion is calculated according to the intensity of pre-edge peaks, and the distortion of the open circuit voltage (OCV) state is set to one (Supplementary Fig. 8b)[29]. The distortion of the pristine and $Li_{1.2}Mn_{0.6}Ni_{0.2}O_{2-\delta}$ samples reaches approximately 195% and 160% when charged to 4.8 V, respectively. When discharged to 2.0 V, the distortion of the pristine sample is still as high as 140%; in sharp contrast, the distortion of the $Li_{1.2}Mn_{0.6}Ni_{0.2}O_2$ sample is almost the same as that in the OCV state. This distortion illustrates that the Mn ions have migrated and stayed in the $Li^+$ layers from the broken rigid $MnO_6$ octahedron, which is caused by the excessive distortion of the pristine sample; hence, the local structure is irreversibly changed. In comparison, the distortion of the $Li_{1.2}Mn_{0.6}Ni_{0.2}O_{2-\delta}$ sample is flexible, and the Mn ions always remain in the $MnO_6$

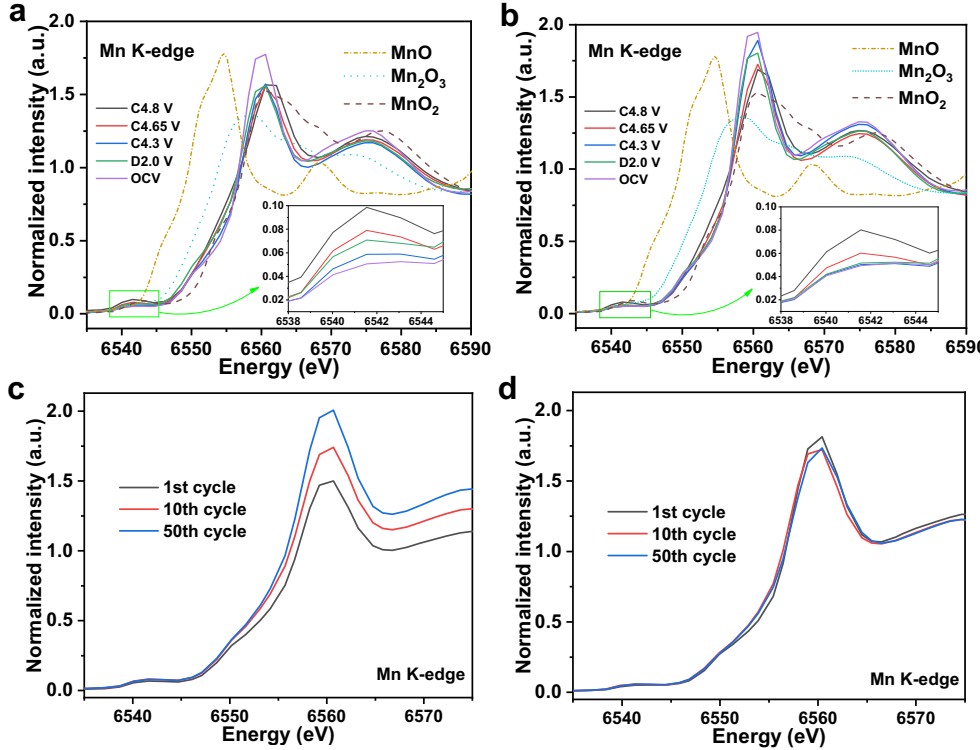

**Fig. 4 Electronic structure of Li-rich Mn oxide-based layered cathode materials.** Ex situ Mn *K*-edge XAS of the pristine (**a**) and M-LRO (**b**) samples at different charge–discharge states; the insets represent the enlarged pre-edge peaks. Ex situ Mn *K*-edge XAS of pristine (**c**) and M-LRO (**d**) recorded after the 1st, 10th and 50th cycles. The term a.u. means arbitrary units.

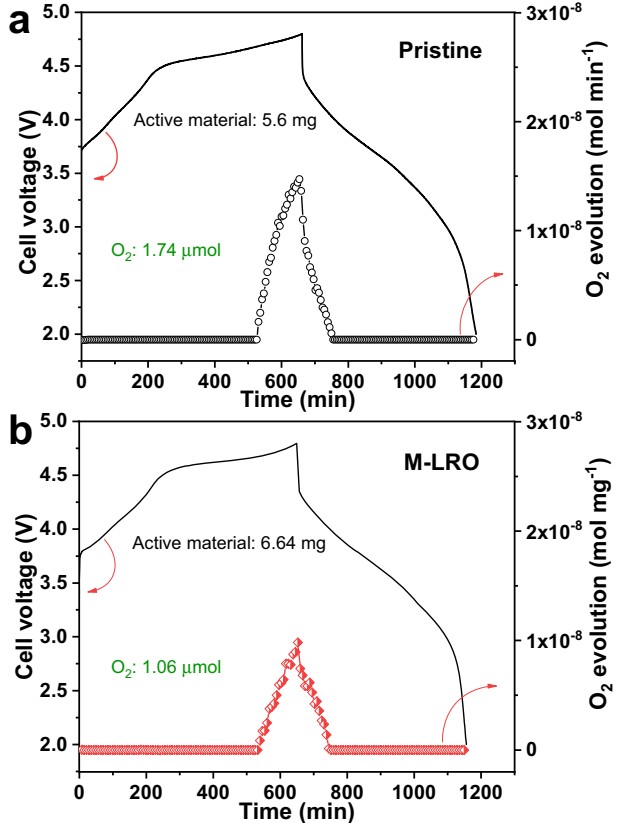

**Fig. 5 Gas release of the pristine and M-LRO samples by operando DEMS.** Initial voltage profiles and oxygen evolution curves of the pristine (**a**) and M-LRO samples (**b**).

octahedral structure. The reversibility of the structural change can also be proven by the Ni *K*-edge spectra at different delithiated states (detailed discussion in Supplementary Fig. 9). The absorption edges of the $Li_{1.2}Mn_{0.6}Ni_{0.2}O_{2-\delta}$ sample discharged to 2.0 V can completely overlap with those of the OCV state, which shows that the distortion of the $NiO_6$ octahedra is completely reversible and reduces $Li^+/Ni^+$ mixing in the $Li_{1.2}Mn_{0.6}Ni_{0.2}O_{2-\delta}$ sample (Supplementary Fig. 9b). Therefore, the voltage decay can be significantly suppressed. To further reveal that the softened structure can inhibit the migration of Mn ions, ex situ Mn *K*-edge spectra of both samples with different cycles are given in Fig. 4c, d. The absorption edges and white line of the pristine sample change continuously from the first cycle to the 50th cycle, which also indicates that the Mn ions have migrated to $Li^+$ layers from the irreversible rigid $MnO_6$ octahedral distortion. The stable Mn *K*-edge spectra of the $Li_{1.2}Mn_{0.6}Ni_{0.2}O_{2-\delta}$ sample illustrate that the softened crystal structure is beneficial to reversible flexible distortion and constrains Mn-ion migration. The almost unchanged Ni *K*-edge spectra of the $Li_{1.2}Mn_{0.6}Ni_{0.2}O_{2-\delta}$ sample are compared with those of the pristine sample from the first to the 50th cycle, illustrating that the softened crystal structure improves the reversible flexible distortion of the $NiO_6$ octahedron and mitigates antisite cation mixing (detailed discussion in Supplementary Fig. 9c, d).

In addition, soft XAS is sensitive to the oxide states and orbital information of different elements. The ex situ Ni and Mn *L*-edge spectra and O *K*-edge spectra of both samples are collected in total electron yield (TEY) mode, which can detect to a depth of 5–10 nm (Supplementary Fig. 10)[31,32]. Regarding the $Li_{1.2}Mn_{0.6}Ni_{0.2}O_{2-\delta}$ sample, a small amount of $Mn^{3+}$ is gradually oxidized to $Mn^{4+}$ when charging to 4.8 V, which means that the Mn ions take part in charge compensation; this agrees with the hard XAS results. The Mn can return to its original state when

discharged to 2.0 V, notably, there is also some $Mn^{3+}$ in the pristine sample, which is caused by the activation of the $Li_2MnO_3$ component and structural rearrangement (Supplementary Fig. 10a, b). The Ni $L_{3,2}$-edges are located at approximately 853 and 870 eV, respectively; in particular, the $L_3$-edge shows asymmetric split peaks at higher and lower energies. We find that the lower energy split peak begins to decrease, while the intensity of the higher energy peak increases when charged to 4.55 V, which illustrates that $Ni^{2+}$ is oxidized to $Ni^{+3/4}$. However, when charge to 4.8 V, the pristine sample shows $Ni^{+4}$; in contrast, the $Li_{1.2}Mn_{0.6}Ni_{0.2}O_{2-\delta}$ sample shows partially reduced $Ni^{4+}$, which is caused by a reductive coupling mechanism (RCM)[31,33]. Specifically, the $O^{2-}$ is oxidized to peroxo-like species, and the electrons are transferred to the LHB closer to the O 2p nonbonding band (Fig. 1c), thus reducing $O_2$ release, as shown in Fig. 5 (the corresponding test device diagram is in Supplementary Fig. 11). This phenomenon can also be observed in the bulk with FY mode (Supplementary Fig. 12). Regarding the O K-edge spectrum, the peaks below 534 eV are ascribed to the hybridization of O 2p with the TM 3d band, while the peaks above this point are attributed to the mixing of O 2p with the 4sp band[34]. The arrows indicate the $e_g$ orbital (Supplementary Fig. 10e, f). The changes in the $e_g$ peaks with increasing voltage indicate structural distortion, and the comparison shows that the distortion is larger in the pristine sample when charged to 4.8 V; moreover, when discharged to 2.0 V, the pristine $e_g$ peak cannot be restored, indicating that the distortion is irreversible due to the irreversible migration of TMs from the damaged rigid octahedral structure (Supplementary Fig. 10e, f and Supplementary Fig. 13). This result further emphasizes the importance of flexible distortion caused by oxygen vacancies.

Suppressed cation migration during charge–discharge can be observed in the HAADF-STEM images (Supplementary Fig. 14). The atomic-resolution image reveals that TMs significantly migrate to Li layers from the octahedral sites of the TM layers for the pristine sample, as shown by the rectangular box in Supplementary Fig. 14a. Moreover, the dark region indicates the existence of nanovoids, which are caused by cation and anion migration and oxygen release accompanied by oxygen redox chemistry; this promotes the degradation of the lattice structure and voltage decay[35]. In contrast, there is almost no change, outside of the slightly twisted TM arrangement, for the $Li_{1.2}Mn_{0.6}Ni_{0.2}O_{2-\delta}$ sample when charged to 4.8 V. This result is consistent with the above analysis. The migrating TMs can be bound in the flexible lattice; thus, distortion is driven by the reduced Mn and oxygen vacancies. When the pristine sample is discharged to 2.0 V, the dumbbell-shaped TM–TM structure becomes blurred (Supplementary Fig. 14c), while some TMs remain in the $Li^+$ layers, which could be associated with the side reaction of oxygen with the electrolyte and the irreversible migration of TMs from the broken rigid $TMO_6$ octahedron. However, the crystal structure of $Li_{1.2}Mn_{0.6}Ni_{0.2}O_{2-\delta}$ sample remains intact because of its reversible distortion and oxygen redox (Supplementary Fig. 14d). Moreover, compared with the retention of the superstructure peaks of the $Li_{1.2}Mn_{0.6}Ni_{0.2}O_{2-\delta}$ sample[36], the disappearance of the pristine sample shows that the TMs migrate irreversibly from the disturbed rigid crystal structure (Supplementary Fig. 15), which is in line with the above results. The softened crystal structure mitigates TM migration by allowing reversible distortion.

**Oxygen redox enhanced by the flexible distortion of O vacancies.** The $Li_{1.2}Mn_{0.6}Ni_{0.2}O_{2-\delta}$ electrochemical energy storage performance is also associated with reduced oxygen release and enhanced oxygen redox chemistry. The amount of oxygen release

was measured by operando DEMS (Fig. 5 and Supplementary Fig. 11). We find that the oxygen production of pristine and $Li_{1.2}Mn_{0.6}Ni_{0.2}O_{2-\delta}$ samples is 0.31 and 0.16 $\mu$mol $mg^{-1}$, respectively. The reduced oxygen production can be attributed to the oxygen vacancies, and it can reduce the parasitic reaction between oxygen and electrolyte; therefore, less corrosion of the cathode material reduces the dissolution and migration of TMs, which can be verified by the thinner CEI (Supplementary Fig. 16). As a result, the voltage fade is strongly mitigated. Although the oxygen release is reduced according to the DEMS results of the $Li_{1.2}Mn_{0.6}Ni_{0.2}O_{2-\delta}$ sample, oxygen redox is enhanced, which can be rationalized by the presence of peroxo-like $(O_2)^{2-}$ species[33,37,38] and molecular oxygen[36,39]. First, the peroxo-like species caused by oxygen redox can be detected in the O 1 s XPS spectra (Supplementary Fig. 17). To detect the bulk structure, samples are etched for different times. When the etching time exceeds 9 min, the peak of the electrolyte oxidation species (533 eV) disappears, illustrating that the influence of the surface electrolyte is removed. In the figures, the purple region (530.5 eV) indicates the existence of lattice oxygen $(O^-/O_2^{2-})$ caused by the oxygen redox chemistry. The larger the area is, the more peroxo/superoxo-like species, therefore, we can find that the oxygen redox activity of the $Li_{1.2}Mn_{0.6}Ni_{0.2}O_{2-\delta}$ sample is stronger, and the different etching times can also demonstrate this enhanced phenomenon in the bulk. Second, the high-resolution O K-edge RIXS spectrum can prove the existence of molecular $O_2$ in Fig. 6a–d. In the low energy loss region, vibrational peaks are observed when both samples are charged to 4.8 V. Specifically, the energy loss at ~0.2 eV corresponds to a vibrational frequency of 1600 $cm^{-1}$, matching the vibration of $O_2$ molecules caused by oxygen redox[36,39,40]. However, we speculate that the more confined change in the inelastic peaks of the $Li_{1.2}Mn_{0.6}Ni_{0.2}O_{2-\delta}$ sample is related to the distortion of octahedral $TMO_6$. The softened crystal structure preserves the octahedral $TMO_6$ structure due to flexible distortion; therefore, the vibration of molecular oxygen will be confined in the undisturbed structure, which can be demonstrated by the redshift of the peaks between $-1.0$ and $-0.2$ eV in Fig. 6e. The peaks disappear when the samples are discharged to 2.0 V. The results show that O-redox occurs in the bulk and is reversible. The oxidation species, namely, $O^{2-}$, O$^-/O_2^{2-}$ and $O_2$, coexist in the $Li_{1.2}Mn_{0.6}Ni_{0.2}O_2$ cathode material due to oxygen redox chemistry. Moreover, the peak located at an energy loss of 8 eV also reinforces the existence of oxygen holes caused by the removal of electrons from the O 2p band[41]. The stronger peak means that the oxygen redox activity is higher for the $Li_{1.2}Mn_{0.6}Ni_{0.2}O_{2-\delta}$ sample (Fig. 6a, c). The oxygen holes can also be confirmed by the O K-edge soft XAS spectrum[42–44]. Figure 7a, b show the ex situ O K-edge soft XAS patterns of the pristine and $Li_{1.2}Mn_{0.6}Ni_{0.2}O_{2-\delta}$ samples in FY mode, respectively. As the dashed line shows, the intensities of the $e_g$ peaks weaken during charging, which indicates that structural distortion is irreversible for the pristine sample during cycling due to the structural arrangement accompanied by oxygen release. In comparison, the modified sample still shows a peak with weak intensity (Supplementary Fig. 13), which means that the softened structure results in flexible distortion to inhibit TM migration and structural arrangement, which is in line with the TEY mode results (Supplementary Fig. 10). The integrated intensity of the O K-edge XAS spectra between 525 and 534 eV demonstrates the distribution of oxygen hole states above the Fermi level[34]. Figure 7c shows the variation in the integrated intensity calculated according to the shaded area in Fig. 7a, b at different charge–discharge states. The results show that the number of oxygen holes gradually increase as charging progresses and reaches a maximum of 4.8 V for both samples. However, the

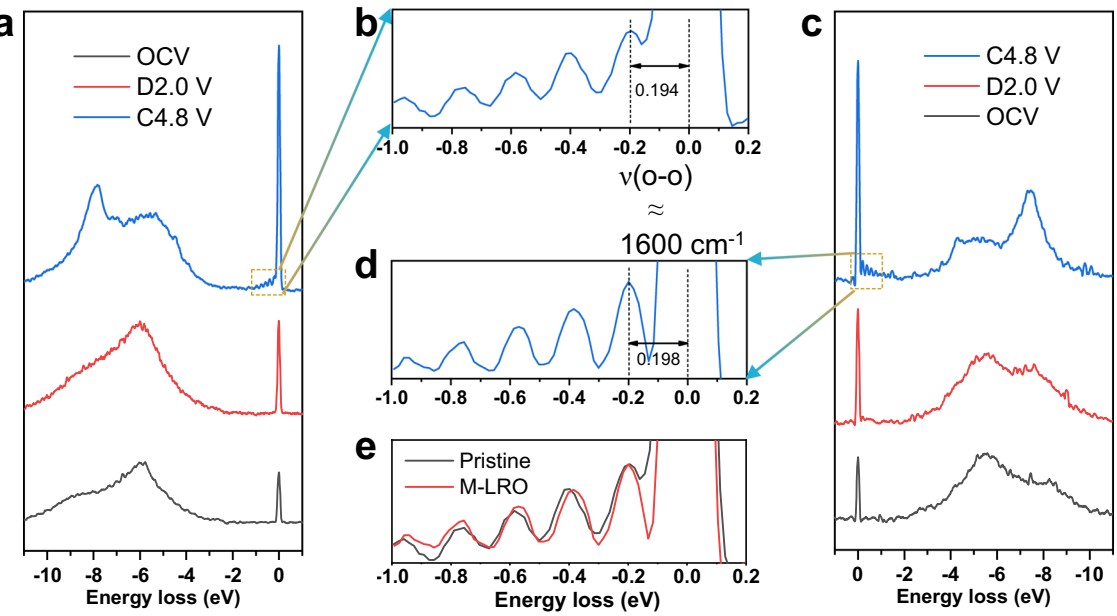

**Fig. 6 Ex situ RIXS evidence for O₂ formation.** O *K*-edge RIXS spectra at an excitation energy of 531 eV for the different charge–discharge states of the pristine sample and the enlarged feature when charged to 4.8 V (**a, b**). O *K*-edge RIXS spectra at an excitation energy of 531 eV for the different charge–discharge states of the M-LRO sample and the enlarged feature when charged to 4.8 V (**c, d**). The energy loss of ~0.2 eV represents the vibration of molecular oxygen. **e** Overlapping pattern of both samples to illustrate the redshift in the M-LRO spectrum.

integrated intensity of $Li_{1.2}Mn_{0.6}Ni_{0.2}O_{2-\delta}$ is greater than that of the pristine sample, meaning that the oxygen redox activity is greater for the modified sample. It is worth noting that, although there are more holes in the modified sample, the emissive electrons from oxygen can be accepted by the (TM-O)* band instead of O₂ release; hence, the structure could be considered as stable[33]. In addition to absorption spectroscopy, the O *K*-edge RIXS map also illustrates the oxygen redox chemistry due to its sensitivity to the oxidation state of oxygen[45]. The two distinct features at the emission energy of 525 eV are attributed to TM(3*d*)-O(2*p*) mixing in Fig. 7d, e. In addition, the feature in the encircled area represents the fingerprint of oxygen redox due to its high resolution to the oxidation state of the oxygen ion[17,46]. This fingerprint feature is more distinguishable in the $Li_{1.2}Mn_{0.6}Ni_{0.2}O_{2-\delta}$ sample charged to 4.8 V than that of the pristine sample, which illustrates that there is more oxygen redox activity in the modified sample. This is in good agreement with the electrochemical energy storage performances and XAS data. To further illustrate the nature of the enhanced reversible oxygen redox due to the oxygen vacancies and reduced Mn, the Mn *L*-edge RIXS of both samples are obtained as shown in Fig. 7f, g. To compare the intensity change, the spectra are normalized to one, which can demonstrate the number of scattering photons or difficulty in scattering these photons. The elastic peaks located at 0 eV energy loss are attributed to the refilling of the 2*p* core holes by the incident electron, the yellow area represents the *d–d* excitation between 1 and 5 eV, which is caused by the refilling of 2*p* core holes by electrons occupying the 3*d* valence or conduction band, and the blue region means the charge transfer (CT) excitation[47–49]. First, the CT intensity is related to the extent of hybridization of the Mn–O bond. Therefore, the hybridization of Mn–O bonds decreases with increasing voltage in both samples. However, the peak intensity of the $Li_{1.2}Mn_{0.6}Ni_{0.2}O_{2-\delta}$ sample can revert back to the original state compared to the pristine sample, which may be caused either by flexible distortion or less oxygen release in the modified sample[50,51]. Then, the peak intensity of the *d–d* transition represents the number of emitted photons; specifically, if the *d–d* Coulombic interaction (repulsion) term *U*

is too large, electrons can be localized in the d atomic orbital. Hence, fewer photons can be detected during de-excitation. The *d–d* transition can be seen at the OCV for both samples, as the dotted line shows; however, the intensity of the transition for the pristine sample is weaker than that for the $Li_{1.2}Mn_{0.6}Ni_{0.2}O_{2-\delta}$ sample, indicating that *U* is smaller in the modified sample. With the extraction of Li⁺, although the repulsion gradually increases in both samples, electrons are more easily removed from the O 2*p* nonbonding band accompanied by oxygen release in the pristine sample, as shown in Fig. 7h, i. When Li⁺ is reinserted into the structure, the *d–d* excitation peak of the $Li_{1.2}Mn_{0.6}Ni_{0.2}O_{2-\delta}$ sample can again revert back to the original intensity at the OCV state; in contrast, the peak of the pristine sample almost disappears in the limit of resolution of the ex situ Mn *L*-edge RIXS spectra, which further illustrates that the MnO₆ octahedral structure is irreversible and destroyed due to rigid distortion.

**Unified picture of enhanced oxygen redox in oxides with oxygen vacancies and reduced TMs.** We use DFT to uncover the higher migration barrier of Mn ions for the $Li_{1.2}Mn_{0.6}Ni_{0.2}O_{2-\delta}$ sample, which can explain why Mn is stable during electrochemical cycling of the $Li_{1.2}Mn_{0.6}Ni_{0.2}O_{2-\delta}$. The calculation models are shown in Supplementary Fig. 18. According to these models, the diffusion channel is given in Supplementary Fig. 19a, b. First, the Mn-ion migration path is more tortuous from the octahedral MnO₆ structure in the TM layers to the tetrahedral LiO₄ structure in the Li layers of the $Li_{1.2}Mn_{0.6}Ni_{0.2}O_{2-\delta}$ sample than that of the pristine sample, meaning that migration is difficult during electrochemical cycling. Quantitatively, the Mn-ion migration barrier of the $Li_{1.2}Mn_{0.6}Ni_{0.2}O_{2-\delta}$ sample is 2.50 eV; in contrast, the migration barrier of the pristine sample is lower (2.23 eV) (Supplementary Fig. 19c), which is in good agreement with the above experimental analysis. In particular, the HAADF-STEM images (Supplementary Fig. 14) show that this result is consistent with the previous calculation[52]. The flexible octahedral MnO₆ structure distortion is also mimicked, and the Mn atom highlighted with a dashed circle close to the oxygen vacancy is used as a simulation target to study the degree of distortion

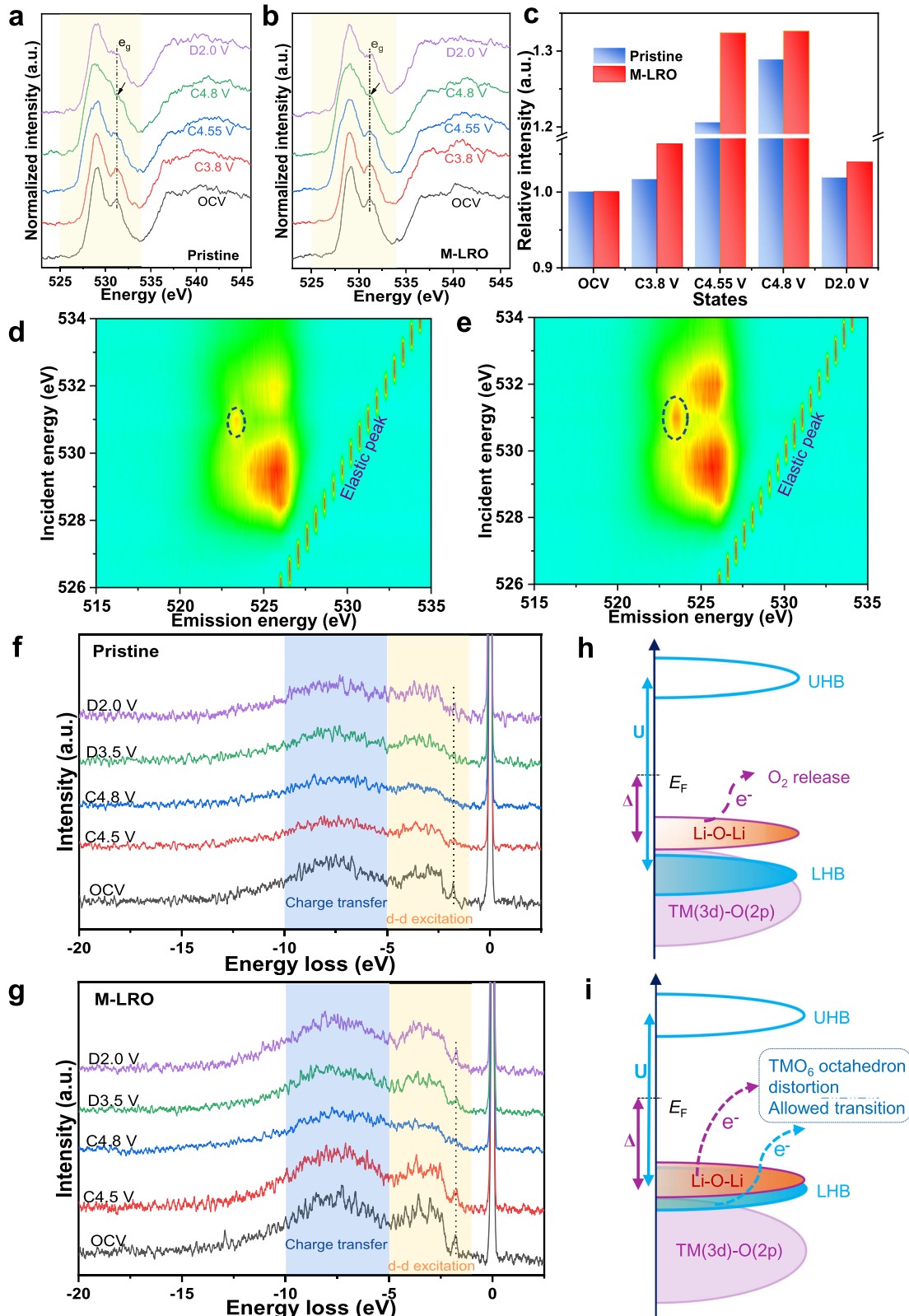

**Fig. 7 Experimental evidence for reversible oxygen redox.** O $K$-edge XAS spectra of the pristine $Li_{1.2}Mn_{0.6}Ni_{0.2}O_2$ (**a**) and $Li_{1.2}Mn_{0.6}Ni_{0.2}O_{2-\delta}$ (M-LRO) (**b**) samples in FY mode at different voltage states. The vertical lines indicate the oxygen $e_g$ orbital, and its strength illustrates the degree of distortion of the crystal structure. The content of oxygen holes can be obtained from the shaded area in the (**a**) and (**b**) patterns, and the corresponding integrated intensity is shown in (**c**). The O $K$-edge RIXS maps of the pristine (**d**) and M-LRO (**e**) samples when charged to 4.8 V are collected with an incident energy between 526 and 534 eV. The ex situ Mn $L$-edge RIXS spectra of pristine (**f**) and M-LRO (**g**) are recorded with at an excitation energy of 643.4 eV. The area shaded in blue and yellow represent the charge transfer and $d$–$d$ transition, respectively. The peak strength of the $d$–$d$ excitation illustrates the size of the $U$ value, which can be qualitatively presented for the **h** pristine and **i** M-LRO samples. The term a.u. means arbitrary units.

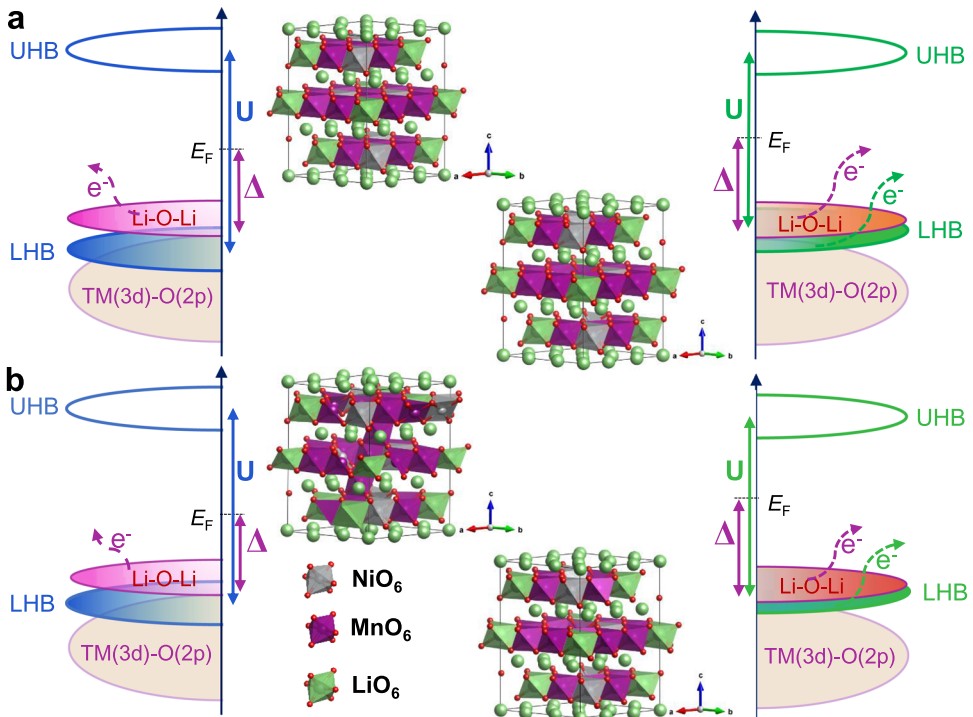

**Fig. 8 Schematics of the band structure.** Mott–Hubbard splitting of the pristine $Li_{1.2}Mn_{0.6}Ni_{0.2}O_2$ (left) and $Li_{1.2}Mn_{0.6}Ni_{0.2}O_{2-\delta}$ (right) samples in the OCV state (**a**) and relithiation state after full delithiation (**b**). In the initial cycle, the $U$ of $Li_{1.2}Mn_{0.6}Ni_{0.2}O_{2-\delta}$ is always smaller than that of the pristine sample, and there is a better reversible overlap between the filled LHB and O $2p$ nonbonding band in the $Li_{1.2}Mn_{0.6}Ni_{0.2}O_{2-\delta}$ sample. The crystal models show the irreversible Mn migration to the Li layer in the pristine sample when discharged to 2.0 V.

(Supplementary Fig. 20)[53]. The results are shown in the histogram, and the distortion of the pristine sample is larger than that of the $Li_{1.2}Mn_{0.6}Ni_{0.2}O_{2-\delta}$ sample with lithium extraction. More significantly, the distortion of the $Li_{1.2}Mn_{0.6}Ni_{0.2}O_{2-\delta}$ sample can almost be restored to the original state; in contrast, the distortion of the pristine sample is still very large during $Li^+$ reinsertion, which is mainly ascribed to the oxygen vacancies softening the crystal structure and preventing damage to the structure caused by rigid distortion, which is in line with the XAS results.

The electronic band structure is evaluated using the density of states (DOS), as calculated by DFT at different states (Supplementary Fig. 21), and the schematized patterns that are combined with the RIXS results are shown in Fig. 8. First, for the originating state, the Mott-Hubbard splitting, namely, the $d$–$d$ Coulombic interaction term $U$, is smaller in the $Li_{1.2}Mn_{0.6}Ni_{0.2}O_{2-\delta}$ sample due to the existence of a small amount of $Mn^{3+}$ and the distortion caused by it. Because $U$ is inversely proportional to the orbital volume, the larger $Mn^{3+}$ radius relative to $Mn^{4+}$ will also result in a smaller $U$[12]. Then, in the state of complete delithiation, the Li-O-Li band is irreversibly damaged. Therefore, the analysis is ignored at this state. However, in terms of qualitative analysis, the Mn ions are oxidized to +4 in both samples, and the smaller radius and half-filled $t_{2g}^3$ states both cause an increase in $U$. At this state, electrons are easily removed from the O $2p$ nonbonding band due to the $O_2$ release in both samples when comparing this state with the other states. The difference is that this trend is more obvious in the pristine sample, which is in line with the DEMS and RIXS results. Last, when $Li^+$ is reinserted back into the structure when discharged to 2.0 V, the crystal structure of the $Li_{1.2}Mn_{0.6}Ni_{0.2}O_{2-\delta}$ sample can recover to the initial state; therefore, $U$ decreases. In comparison, the broken structure prevents the decrease in term $U$ for the pristine sample, for instance, irreversible Mn migration to $Li^+$ layers along with oxygen release, which is in good agreement with Mn $L$-edge RIXS

results. Moreover, the reduced $d$ band width and the $d$ band centre being closer to the Fermi level, which is due to oxygen vacancies, facilitate the adjustment of $U$ and $\Delta$. (Supplementary Fig. 21). Therefore, the calculation results and the experimental results, especially the RIXS results, reveal why the reduced TMs (Mn) and oxygen vacancies can improve performance.

## Discussion

With advanced characterization techniques, we studied the inherent nature of enhanced reversible oxygen redox caused by reduced Mn and oxygen vacancies. On the one hand, the reduced Mn can take part in charge compensation; on the other hand, oxygen vacancies can not only decrease $O_2$ release and mitigate Mn migration but can tolerate the structural distortion that occurs during electrochemical cycling by allowing flexible distortion. Therefore, the voltage decay is significantly suppressed. Specifically, the $d$–$d$ Coulombic interaction term $U$ can be decreased by the introduction of reduced Mn and oxygen vacancies, which can adjust $U$ and prevent it from being much larger than $\Delta$; thus, oxygen release can be decreased and reversible anion redox chemistry can be enhanced. Furthermore, flexible distortion can prevent the irreversible migration of TMs and maintain the integrity of the crystal structure; therefore, $U$ can reversibly return to its initial state. As a result, the enhanced O redox chemistry becomes reversible. In summary, this work demonstrates an interesting approach to mitigate voltage decay and enhance reversible oxygen redox chemistry in Li-based cathode active materials by chemical reduction strategies and defect engineering.

## Methods

**Materials synthesis.** The precursor $Ni_{0.25}Mn_{0.75}CO_3$ was synthesized by the coprecipitation method. First, a certain amount of $MnSO_4 \cdot 5H_2O$ and $NiSO_4 \cdot 6H_2O$ were dissolved into 100 ml of deionized water, and then 50 ml of $Na_2CO_3$ (2 M) was slowly added to the above mixed solution and stirred for 12 h. Finally, the stirred solution was washed three times with deionized water and twice with ethanol and placed in an oven to dry at 80 °C. To obtain the pristine sample, an

excess of 5% LiOH·H$_2$O was mixed with the corresponding proportion of the precursor. Then, the mixture was transferred to a crucible and heated to 500 °C at a rate of 2 °C/min in a muffle furnace, kept isothermal for 300 min, heated at 5 °C/min to 900 °C for 720 min, and finally naturally cooled to room temperature. For the modified Li$_{1.2}$Mn$_{0.6}$Ni$_{0.2}$O$_{2-\delta}$ sample, the synthesized pristine sample was dispersed in different concentration of hydrazine hydrate solutions and stirred for different times. Then, the mixed solution was washed twice with deionized water by centrifugation and twice with ethanol. Finally, the sample was dried in an oven at 80 °C and then dried in a vacuum oven at 120 °C for 12 h.

**Electrochemical measurements**. The prepared material was mixed with polyvinylidene fluoride (PVDF, Arkema, French) and carbon black (Lion Specialty Chemicals Co., Ltd) at a weight ratio of 80%:10%:10%, the mixture was ground with agate mortar for 10 ± 2 min in an air environment, and N-methyl pyrrolidinone (Shanghai Macklin Biochemical Co., Ltd.) was added during the grinding process. The obtained slurry was coated on aluminium foil (Shenzhen Kejing Star Technology Co., Shenzhen) at a thickness of 16 μm and with a purity of more than 99.35%. Next, the sample was dried for 12 h under vacuum at 120 °C. The active material mass, diameter and thickness of the positive electrode were 2–3 mg/cm$^2$, 9 mm and 20–30 μm, respectively. Lithium metal with a purity greater than 99.9% was used as the counter and reference electrode, and its thickness and diameter was 0.45 mm and 15.6 mm, respectively (China Energy Lithium Co., Ltd. Tianjin). The assembly and disassembly of the CR2025-type coin cells were carried out in a glove box filled with argon gas and the content of H$_2$O and O$_2$ is less than 0.1 ppm (Etelux Inertia Gas System (Beijing) Co., Ltd.). A Whatman GF/D glass microfibre filter was used as the separator. A high-voltage electrolyte was purchased from Beijing Institute of Chemical Reagents, and the typical electrolyte formula was LiPF$_6$ (1 M) in a 1:1 volume ratio of fluoroethylene carbonate and dimethyl carbonate. The amount of electrolyte for each cell was 60–80 μL. PITT was performed with an electrochemical workstation (Metrohm-Autolab, PGSTAT 302N). The sample was gradually charged from OCV to 4.8 V with a voltage step of 0.1 V. During this process, charging occurred at a constant voltage for 1800 s, and was then relaxed for the same amount of time to obtain the chronoamperometric curve (I–t). Galvanostatic charge–discharge was performed from 2 to 4.8 V at specific currents with a battery tester (NEWARE CT-4000) in an environmental chamber (22 ± 3 °C).

**Physicochemical characterizations**. The crystal structures of materials were characterized by performing powder XRD with a Smartlab XRD (Rigaku, a CuKα radiation source) in a 2 theta range of 10–80°. The in situ XRD electrode was assembled in a cell mould with a beryllium window (Beijing Scistar Technology Co., Ltd), the positive electrode without a current collector is cut into a 1 × 1 cm$^2$ square, and the counter electrode and separator were the same as the coin cells. The specific current was 25 mA/g for the in situ XRD cell, and data were collected every 15 min. Ex situ TEM (Tecnai G2 F30) was used to observe the CEI and morphology after cycling. Ex situ spherical aberration-corrected transmission electron microscopy (JEM ARM200F, JEOL, Tokyo, Japan) was used to observe the atomic structure of materials at different states. Ex situ XPS with a monochromatic X-ray source (Al Kα) is used to detect the chemical state and electronic state of the material in different states. Ex situ $^7$Li NMR spectra were collected by a solid-state NMR Spectrometer (Bruker AVANCE III, 400 MHz) under MAS rate of 12.8 kHz. For the ex situ measurement, the cells were disassembled, washed with dimethyl carbonate, packaged in aluminium-plastic bags in an Ar-filled glove box, and then brought to the testing site for rapid sampling loading and testing. The overall exposure time in air is less than 2 min. Oxygen production was recorded by operando DEMS (Linglu Instruments Co., Ltd. Shanghai). The DEMS cell was assembled with a Swagelok-type cell, where the diameter and thickness of electrode disc were 12–19 mm and 30–50 μm, respectively. The assembled cell was connected to the gas path of the mass spectrometer (Pfeiffer, OminiStar GSD 320). Note that the total carrier gas (Ar) was 3 mL/min, and the flow was 0.5 mL/min through the Swagelok cell. The cell was continuously ventilated for 10-15 h until the baseline was stable and then charged and discharged at a specific current of 25 mA/g. The overall operating diagram and Swagelok cell are shown in Supplementary Fig. 11a, b.

**Synchrotron radiation characterization**. Hard XAS was conducted at the KMC2 beamline of the synchrotron BESSY II at Helmholtz-Zentrum Berlin für Materialien und Energie (HZB, Germany)[54]. O K-edge soft XAS was performed at the Russian–German Beamline in the TEY mode at HZB, the O K-edge spectra in the FY mode were collected at beamline BL08U1-A in the Shanghai Synchrotron Radiation Facility (SSRF), and all other TM L-edge spectra in the TEY and FY modes were tested at beamline 20A1 of the Taiwan Light Source (TLS) at HsinChu, Taiwan, China. The RIXS maps and spectra of O and Mn were recorded at the U41-PEAXIS beamline at HZB[55,56].

**Neutron powder diffraction**. Data was obtained at the Vulcan beamline spallation neutron source (SNS) at Oak Ridge National Laboratory[57]. The size of the incident beam was 5 mm × 12 mm, the diameter of the receiving collimator was 5 mm, and the bandwidth of the incident electron beam was 0.7–3.5 Å. Double-disk choppers at a speed of 30 Hz were employed, and a 0.5–2.5 Å d-spacing was allowed in the diffraction

pattern of the θ ± 90 detector banks. In high resolution mode, Δd/d was ~0.25%. The power of SNS was 1400 kW. At a temperature of 25 °C, NPD data acquisition took 3 h, and VDRIVE software was used to reduce the data. A full-pattern Rietveld refinement was implemented using the GSAS program with an EXPGUI interface[58].

**DFT calculation**. We used the hexagonal Li$_3$Mn$_3$O$_6$ structure with the R-3m space group as the initial structure[59]. We built a 3 × 3 superstructure of Li$_3$Mn$_3$O$_6$ and then obtained the Li$_{27}$Mn$_{27}$O$_{54}$ structure. To obtain the Li$_{1.2}$Mn$_{0.6}$Ni$_{0.2}$O$_2$ composition, we randomly replaced five Mn atoms with Li and six Mn atoms with Ni, resulting in the Li$_{32}$Mn$_{16}$Ni$_6$O$_{54}$ (equivalent to Li$_{1.19}$Mn$_{0.59}$Ni$_{0.22}$O$_2$) composition. For the Li$_{1.2}$Mn$_{0.6}$Ni$_{0.2}$O$_{2-x}$ structure with O vacancies, we omitted three O atoms in the Li$_{32}$Mn$_{16}$Ni$_6$O$_{54}$ system and obtained the Li$_{32}$Mn$_{16}$Ni$_6$O$_{51}$ structure, which is equivalent to Li$_{1.19}$Mn$_{0.59}$Ni$_{0.22}$O$_{1.89}$. First-principles calculations were performed by density functional theory (DFT) using the Vienna Ab-initio Simulation Package (VASP) package[60]. The generalized gradient approximation (GGA) with the Perdew-Burke-Ernzerhof (PBE) functional was used to describe the electronic exchange and correlation effects[61–63]. Uniform G-centred k-point meshes at a resolution of $2\pi \times 0.04$ Å$^{-1}$ and Methfessel-Paxton electronic smearing were adopted for integration in the Brillouin zone for geometric optimization. The simulation was run with a cut-off energy of 500 eV throughout the computations. These settings ensured convergence of the total energies to within 1 meV per atom. Structural relaxation proceeded until all forces on the atoms were less than 1 meV Å$^{-1}$ and the total stress tensor was within 0.01 GPa of the target value. To describe the on-site Coulombic interaction, the DFT + U approach was used to calculate all the elementary reaction steps, and the U terms of Mn-3$d$ and Ni-3$d$ were 3.9 and 6.4 eV, respectively. The octahedral MnO$_6$ distortion ($\triangle$) was calculated by the equation $\triangle = \frac{1}{6}\sum\left(\frac{R_i - R_{av}}{R_{av}}\right)^2$, where $R_{av}$ is the average bond length and $R_i$ is the individual bond length of Mn−O in the octahedral MnO$_6$. The migration barriers of Mn were calculated using the climbing image nudged elastic band (ciNEB) method[64].

**Reporting summary**. Further information on research design is available in the Nature Research Reporting Summary linked to this article.

## Data availability

The data that support the findings of this study are available from the corresponding author upon reasonable request. The curves and graphs of the data generated in this study have been deposited in the figshare database under accession code https://figshare.com/10.6084/m9.figshare.18093398. Source data are provided with this paper.

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

## Acknowledgements

This work was supported by the International Partnership Program (grant no. 211211KYSB20170060) of the Chinese Academy of Sciences, the National Natural Science Foundation of China (grant nos. 11975238 and 11575192), the Scientific Instrument Developing Project (grant no. ZDKYYQ20170001), and the Strategic Priority Research Program (grant no. XDB28000000) of the Chinese Academy of Sciences. This work was also supported by the Fundamental Research Funds for the Central Universities. The neutron experiments at the SNS user facilities (VULCAN beamline) were sponsored by the Office of Basic Energy Sciences (BES) and the Office of Science of the U.S. DOE. The authors thank Dr. Dmitry Smirnov at the Russian–German Beamline at BESSY-II, HZB, Germany. The authors also thank the staff at beamline 20A1 of the TLS at HsinChu. The authors thank Dr. Hui Fu and the Analytical Instrumentation Centre of Peking University for help with ssNMR testing and analysis and Dr. Jicheng Zhang and Weijin Kong from the University of Chinese Academy of Sciences for help with article revision.

## Author contributions

X.L. proposed and supervised this project; Q.L. synthesized the materials, analyzed data, and wrote the manuscript; D.N. and G.S. performed the hard XAS; D.W. collected the

RIXS data; K.A. characterized the NPD; D.Z., Z.C., and N.Z. tested the soft XAS; and Y.T. gave advice on the manuscript.

## Competing interests

The authors declare no competing interests.
