## [Peer Review File · Nature Communications]

REVIEWER COMMENTS

Reviewer #1 (Remarks to the Author):

In the manuscript entitled "Tuning d-d Coulomb interaction for reversible oxygen redox in Li-rich layered cathode materials" the authors propose to create Oxygen vacancies in order to tune the reversibility of the anionic redox process in Li rich Manganese oxides.

Overall the manuscript is well written, pleasant to read and well supported by a large panel of characterization. As a results the authors clearly demonstrate that the strategy they propose works to reduce O₂ release during the anionic redox process and enhance the reversibility. To my point of view it can be of interest for Li-ion battery community. However several points might be addressed

1) Please add in the main manuscript the corresponding chemical formula associated to LRMO and M-LRO. It is really frustrating and unpleasant that one need to look for the method part in order to guess the chemical formula of the studied materials.

2) In the XRD diffraction spectra, the authors show superstructure peaks around 20°. Can it be associated to an order of O vacancies or to an order of Mn³⁺ which are Jahn-Teller distorted ? This might be important for the DFT results that have been done only on one special O vacancies distribution. Can the authors comment the validity of the DFT results regards to the formation of O-vacancies order and the occurrence of highly JT MnO₆ octahedra associated with Mn³⁺?

3) I somehow disagree with the following sentence in the manuscript "In general, the U is smaller due to the reduced Mn and oxygen vacancy corresponding to a smaller d center band value,...". As a counter example, if one consider a tight-binding model or Hubbard model with U=0 (non correlated), by introducing vacancies, I reduce automatically the bandwidth and I obtain a smaller d-band center shift respect to the Fermi level. If I believe the authors that the introduction of O vacancies mitigate the U versus Delta competition, I am more convinced that this is the effect of reducing the d band width than to reduce the U value. Moreover the U value being an intra-orbital quantity, I do not see by which process O vacancies impact it.

Reviewer #2 (Remarks to the Author):

In this work, the authors present a new approach allowing to mitigate the voltage decay and enhance the reversible oxygen redox chemistry in high energy cathode materials. The study of Li-rich layered oxides cathodes is of significant interest to the battery community due to their high specific capacities. However, the large voltage decay upon cycling does not allow their implementation in practical Li-ion cells. The novelty of their work, compared to previous work in this area, is the adjustment of d-d Coulomb interaction by the reduction of Mn and the incorporation of oxygen vacancy in the Li-rich Mn-based oxides (LRMO) cathode materials to suppress the voltage decay.

However, there are two reasons why I do not see the manuscript fit for publication in Nature Communications, 1) the data analysis is superficial and does not allow a non-specialist to understand some results presented (see detailed comment below) and 2) some conclusions are premature and without clear discussions.

Comment1

The authors present NMR spectra of the pristine and M-LRO samples in supplementary Figure 3. Were these spectra recorded using the magic angle spinning MAS? If it's the case, could the authors replace ⁷Li NMR spectra by ⁷Li MAS NMR spectra and indicate more experimental details (pulse sequence, ...)? I want to add that the peak centered at 0ppm is due to ⁷Li in a diamagnetic environment, which can be LiOH, Li₂CO₃, the data analysis is not sufficient to only correlated it to Li₂CO₃ as reported by the authors.

Also, the authors chose to perform NMR measurements of the 7Li to provide evidence of oxygen vacancy. On page 7, line 132, the authors state: "The oxygen vacancy can be illustrated by the widened sideband pattern in M-LRO sample". This data analysis is rather superficial; what proof do the authors have?

Comment 2

In the data analysis shown in Figure 4 a) and b) (on page 17, lines 347-353), the authors compare the eg peak intensity from the ex-situ O K-edge soft XAS for the both samples (pristine and M-LRO). However, the current form of spectra does not allow the reader to properly observe eg peak differences between the both samples. Could the authors apply a more thorough data analysis? I suggest to use two figures showing the spectra of the pristine and M-LRO samples overlapped, for the first at 4.55V and for the second one at 4.88V (between 525eV and 534eV). Also I want to add that the Figure 4c) is not sufficient to demonstrate eg peak variations.

Comment 3

In Supplementary Figure 8, the PITT spectra of both samples during charging process are presented. This data analysis is superficial. The authors need to clearly discuss the result, especially, the higher value obtained for the M-LRO sample near 2.3V, and the Li diffusion coefficient variation for both samples which are different. The authors state: "electrochemical performance is in good agreement with the result of the PITT". What is the reason for this in regard to this figure?

Comment 4

On page 5, line 83/84, the authors state "It is worth noting that the LMRO with oxygen vacancies and/or spinel phase can improve the properties of cathode materials". LMRO is not defined in the main text. Could the authors specify this term? If it is the abbreviation of Li-rich Mn-based oxides, modify LMRO by LRMO in the whole manuscript.

Comment 5

In the data analysis shown in Figure 4f) and h) (on pages 18-19), the authors state: "the peak intensity of M-LRO sample can revert back to the original state compared to the pristine sample" (lines 382-383), and "the d-d excitation peak of M-LRO sample can again revert back to the original intensity at OCV state, in contrast, the peak of pristine sample almost disappeared" (lines 394-396) I highly recommend to add "in the limit of the resolution of the ex situ Mn L-edge RIXS spectra" because, the pristine spectrum obtained at 2.0V presents a low signal to noise ratio.

Comment 6

On page 9, line 184/185, the authors state: "To reveal the mechanisms of the mitigated voltage decay and increased capacity. In-situ XRD is conducted to monitor the crystal structure change in real time". I suggest to replace "capacity. In-situ" by "capacity, in-situ".

Comment 7

On page 18, lines 371-373, the authors state: "To further illustrate the nature of enhanced reversible oxygen redox due to the oxygen vacancy and reduced Mn. The Mn L-edge RIXS of both samples are collected as Fig. 4f, h shown". I suggest to replace "reduced Mn. The Mn" by "reduced Mn, the Mn".

Reviewer #3 (Remarks to the Author):

Non-hysteretic anionic redox reactions are important for designing high energy density cathode materials for lithium ion batteries. Authors reports on the reversible oxygen capacity in lithium rich LiMnO₂ cathode materials. Although the substitute of TM and O atom enhanced anion redox, those strategies have issues for practical usages, whereas the oxygen vacancy controlling d-d Coulomb

interaction can improve the anion redox reversibility cost-effectively. This work attempted to demonstrate the fundamental understands of the oxygen redox. However, there are contradictions between this work and those published in the literature. It should be further clarified carefully.

1. The authors conclude that oxygen vacancy can enhance the anionic redox and mitigate the TM migration in this study. In Ref 17, the synthesise method and strategies are almost similar.
2. The oxygen vacancy breaks the band of Li-O-Li, which reduces O₂ release. However, the oxygen defect can also effect on d-band sift of both Mn and Ni. In this study, authors discuss the electronic structures of Mn by experiments. The effects of d-band shift of Ni needs to be explained.
3. The thermodynamic instability of oxygen defect would be expected. In this study, the oxygen vacancy reached about 5% in the sample. As is well-known, the defect of oxygen is related to crystal structure collapse. Authors can explain more details why the crystal changes in the modified sample is smaller than it in the pristine sample during lithiation? And the
4. If the oxygen vacancies are generated more (>5%), does the electronic state of anion have similar energy state of TM?

Point-by-point response to the comments

Response to Reviewer #1:

General Comment: In the manuscript entitled "Tuning d–d Coulomb interaction for reversible oxygen redox in Li-rich layered cathode materials" the authors propose to create Oxygen vacancies in order to tune the reversibility of the anionic redox process in Li rich Manganese oxides.

Overall the manuscript is well written, pleasant to read and well supported by a large panel of characterization. As a results the authors clearly demonstrate that the strategy they propose works to reduce O₂ release during the anionic redox process and enhance the reversibility. To my point of view it can be of interest for Li-ion battery community. However several points might be addressed.

Reply: *We would like to sincerely thank the reviewer for the positive comments. We will answer the following questions one by one.*

Specific Comment No. 1: Please add in the main manuscript the corresponding chemical formula associated to LRMO and M-LRO. It is really frustrating and unpleasant that one need to look for the method part in order to guess the chemical formula of the studied materials.

Reply: *Thanks a lot for the reminder. We have replaced all abbreviations of LRMO with $\text{Li}_{1.2}\text{Mn}_{0.6}\text{Ni}_{0.2}\text{O}_2$ in the main manuscript and replaced all abbreviations of M-LRO with $\text{Li}_{1.2}\text{Mn}_{0.6}\text{Ni}_{0.2}\text{O}_{2-\delta}$. The value of δ is 0.05 according to the result of neutron powder diffraction refinement. In addition, we have kept the corresponding abbreviations in the caption part, but where the abbreviation first appeared, the chemical formula was marked. The corresponding replacements have been highlighted throughout the revised manuscript.*

Specific Comment No. 2: In the XRD diffraction spectra, the authors show

superstructure peaks around 20° . Can it be associated to an order of O vacancies or to an order of Mn^{3+} which are Jahn-Teller distorted? This might be important for the DFT results that have been done only on one special O vacancies distribution. Can the authors comment the validity of the DFT results regards to the formation of O-vacancies order and the occurrence of highly JT MnO_6 octahedra associated with Mn^{3+} ?

Reply: Thanks a lot. This is really a good question. We have reviewed lots of literatures, and we found that there is no specific report on the relationship between oxygen vacancy/ Mn^{3+} and superlattice diffraction peaks. According to our observation, when the content of oxygen vacancies is $\sim 5\%$, they have no obvious change on the superlattice peaks, which can be seen in the normalized XRD as shown in Supplementary Figure 1c. In addition, we simulated the XRD diffraction peaks of different oxygen vacancies through calculations as shown in Supplementary Figure 1d below, we can clearly find the diffraction peaks with oxygen vacancy of 0% and 5.6% have a negligible change in intensity and a negligible shift in 2θ of 0.04 degree, which are consistent with the experimental results. However, when the concentration of oxygen vacancies reached 11.1%, the diffraction peaks shifted significantly, which may come from the destruction of the crystal structure. In summary, the order of oxygen vacancies and the order of reduced Mn^{3+} do not have an obvious effect on the superlattice peaks in this material. Note that this issue is indeed worthy of in-depth and systematic research.

Supplementary Figure 1. (a) The experimental XRD patterns of $\text{Li}_{1.2}\text{Mn}_{0.6}\text{Ni}_{0.2}\text{O}_2$ and $\text{Li}_{1.2}\text{Mn}_{0.6}\text{Ni}_{0.2}\text{O}_{2-\delta}$. (b) The simulated XRD patterns of Li -rich oxides with

different oxygen vacancies.

In order to better simulate our experimental results, we created a crystal model containing 54 oxygen atoms. In the model of oxygen vacancies, we randomly delete one oxygen atom from each layer of the crystal structure as shown Supplementary Figure 20. So there are three oxygen vacancies in total rather one vacancy, the total concentration of oxygen vacancies is $3/54 \times 100\% = 5.6\%$, which is similar to the experimental result of neutron refinement.

Supplementary Figure 20. The DFT calculation models. (a) Pristine sample ($\text{Li}_{1.2}\text{Mn}_{0.6}\text{Ni}_{0.2}\text{O}_2$) and (b) M-LRO sample ($\text{Li}_{1.2}\text{Mn}_{0.6}\text{Ni}_{0.2}\text{O}_{2-\delta}$).

In this research, the oxygen redox chemistry has been demonstrated using RIXS and XPS to detect the presence of O_2 and peroxide O_2^{2-} . According to the previous study, the peroxo O-O bond is easier form in the MnO_6 distortion of the straight-type $\text{R}\bar{3}\text{m}$ LiMnO_2 structure (Adv. Mater. 2018, 30, 1705197), therefore, the Li-rich model established based on this structure can better illustrate the oxygen redox chemistry, especially formation of peroxo O-O bond produced by the oxygen redox reaction. Similar to the modeling method in the above article, we also choose LiMnO_2 as the initial structure, and then Li and Ni replace Mn to obtain the final Li-rich structure similar to $\text{Li}_{1.2}\text{Mn}_{0.6}\text{Ni}_{0.2}\text{O}_2$. Based on this model, although we did not redundantly recalculate the formation of peroxo O-O bonds (they has been proved through XPS), we found the distortion of MnO_6 octahedral similar to the above literature. These results are in good agreement with the experiment.

For the role of oxygen vacancies in Li-rich, Cho et al. conducted a detailed study through DFT (*ChemElectroChem* 2016, 3(6): 943-949). In their research, due to the introduction of oxygen vacancies, part of the valence of Mn is reduced ($\text{Li}_2\text{MnO}_{3-\delta}$), and Mn undergoes a redox reaction to participate in charge compensation, which has been verified by the PDOS and the average net charge of Mn. In addition, the migration barrier of Mn ions is calculated, the specific barrier value and migration path are shown below. As the result shown, the migration barrier is higher for the sample with oxygen vacancies $\text{Li}_2\text{MnO}_{3-\delta}$, which would mitigate the phase transformation. According to their analysis, Mn atoms at the intermediate migration states have a coordination number of three, which is insufficient for the formation of a stable tetrahedral or octahedral coordination geometry, therefore, kinetic migration barrier is increased and the phase transformation is suppressed. In our research, there are also oxygen vacancies and reduced Mn. Our calculation result of Mn migration barrier is similar with Cho's. Moreover, in the Meng's research (*Nat. Commun.* 2016, 7, 12108), the oxygen vacancies are introduced in $\text{Li}[\text{Li}_{0.144}\text{Ni}_{0.136}\text{Co}_{0.136}\text{Mn}_{0.544}]\text{O}_2$ by gas-solid interfacial method, it has been studied that the increased Li ions diffusion kinetics and compensation of Mn due to the introduction of oxygen vacancies. All of these DFT results are consistent with our calculations and experimental results. The corresponding modification and citations has been added and highlighted in the Page 6, 20 and 24 in the revised manuscript and the Supplementary Figure 1.

Mn atom migration barriers with respect to migration paths in $\text{Li}_{1.75}\text{MnO}_3$ and

Specific Comment No. 3: I somehow disagree with the following sentence in the manuscript "In general, the U is smaller due to the reduced Mn and oxygen vacancy corresponding to a smaller d center band value,...". As a counter example, if one consider a tight-binding model or Hubbard model with $U=0$ (non correlated), by introducing vacancies, I reduce automatically the bandwidth and I obtain a smaller d-band center shift respect to the Fermi level. If I believe the authors that the introduction of O vacancies mitigate the U versus Δ competition, I am more convinced that this is the effect of reducing the d band width than to reduce the U value. Moreover the U value being an intra-orbital quantity, I do not see by which process O vacancies impact it.

Reply: Thanks a lot for your suggestion. The expression in this sentence is indeed prone to ambiguity. Therefore, we sincerely accept the reviewer's suggestion and revised the statements. In this work, the U mainly comes from the qualitative discussion of the Mn L-edge RIXS rather than the " U " correction of calculation, because " U " correction is fixed throughout the calculation process. As we discussed in the main text, the U term trends to favour on-site localized electrons, and which is inversely proportional to the orbital volume (Nat. Energy, 2018, 3(5), 373-386). First, before charging, part of Mn^{4+} is reduced to Mn^{3+} with a larger radius due to the introduction of oxygen vacancies in the modified sample, so the corresponding U value is smaller and the electrons are more delocalized. This result is reflected in the spectrum that the electrons are easier to be excited and the peak intensity is stronger. Then, at high voltage (charged to 4.8 V), both samples are Mn^{4+} with smaller volume, so the U is larger, and the electrons are more localized and the corresponding RIXS peak intensity is weaker as shown Figure 4f and h. At the same time, the Mn ions are easier to migrate and dissolve in the unmodified sample, plus the more serious oxygen release destroys the coordination between Mn-O, which makes the d-d Coulomb repulsion irreversible

change after discharge (2.0 V), showing an irreversible peak intensity changes. In addition, we very much agree with the reviewer's statement, the oxygen vacancies reduce the d band width, which is responsible for the adjustment of U and Δ . The modification is highlighted in Page 21 of the revised manuscript.

Figure 4f and h. The ex situ Mn L-edge RIXS spectra of pristine (f) and M-LRO (h) are recorded with the 643.4 eV excitation energy.

Response to Reviewer #2

General comment: In this work, the authors present a new approach allowing to mitigate the voltage decay and enhance the reversible oxygen redox chemistry in high energy cathode materials. The study of Li-rich layered oxides cathodes is of significant interest to the battery community due to their high specific capacities. However, the large voltage decay upon cycling does not allow their implementation in practical Li-ion cells. The novelty of their work, compared to previous work in this area, is the adjustment of d–d Coulomb interaction by the reduction of Mn and the incorporation of oxygen vacancy in the Li-rich Mn-based oxides (LRMO) cathode materials to suppress the voltage decay.

However, there are two reasons why I do not see the manuscript fit for publication in Nature Communications, 1) the data analysis is superficial and does not allow a non-specialist to understand some results presented (see detailed comment below) and 2) some conclusions are premature and without clear discussions.

Reply: *We would like to thank reviewer for the comments, which are of great help to the improvement of the content of this article. We will revise and answer the questions according to the reviewer's comments one by one.*

Specific Comment No. 1: The authors present NMR spectra of the pristine and M-LRO samples in supplementary Figure 3. Were these spectra recorded using the magic angle spinning MAS? If it's the case, could the authors replace ^7Li NMR spectra by ^7Li MAS NMR spectra and indicate more experimental details (pulse sequence, ...)? I want to add that the peak centered at 0ppm is due to ^7Li in a diamagnetic environment, which can be LiOH, Li_2CO_3 , the data analysis is not sufficient to only correlated it to Li_2CO_3 as reported by the authors.

Also, the authors chose perform NMR measurements of the ^7Li to provide evidence of oxygen vacancy. On page 7, line 132, the authors state: "The oxygen vacancy can be illustrated by the widened sideband pattern in M-LRO sample". This data analysis is

rather superficial; what proof do the authors have?

Reply: Thanks a lot for the reminder. The ssNMR is used in this paper to prove the existence of oxygen vacancy as an auxiliary method and ^7Li NMR spectra are collected by a Solid-State NMR Spectrometer (Bruker AVANCE III, 400 MHz) under MAS rate of 12.8 kHz. And we have added detailed test method to the experimental part in Page 23.

Then, we agree with the reviewer's statement, the peak centered at 0 ppm is caused by the LiOH, Li_2CO_3 and Li_2O etc. (J. Phys. Chem. C 2018, 122, 3773–3779). These compounds can be found on the surface of Li-rich cathode oxides. However, what we want to express in this sentence is that the peak is mainly caused by Li_2CO_3 , because Li_2CO_3 is the main residual component on the surface of Li-rich cathodes, which can be verified in previous reported papers (ACS Appl. Mater. Interfaces 2019, 11, 11518–11526, J. Power Sources 216 (2012) 179-186, Chem. Mater. 2019, 31, 2545–2554). Therefore, we revised the corresponding statement in the revised manuscript to avoid ambiguity.

After analyzing and discussing with an expert, we think that when there are oxygen vacancies, it will affect the asymmetric distribution of the electron cloud around the nucleus, and the local magnetic field around the nucleus would be perturbed, thereby affecting the anisotropy of the chemical shift of the nucleus, which will broaden the spectrum peak (Solid State Ionics 2012, 225, 488–492, Acta Phys. -Chim. Sin. 2020, 36 (4), 1902019). We think this is the reason for widened sideband pattern in the $\text{Li}_{1.2}\text{Mn}_{0.6}\text{Ni}_{0.2}\text{O}_{2-\delta}$ sample, and the corresponding explanation and acknowledge have been added in the revised manuscript (Please see Highlights in Page 7, 23 and 24 of the revised manuscript).

Specific Comment No. 2: In the data analysis shown in Figure 4 a) and b) (on page 17, lines 347-353), the authors compare the e_g peak intensity from the ex-situ O K-edge soft XAS for the both samples (pristine and M-LRO). However, the current form of spectra does not allow to the reader to properly observe e_g peak differences between the

both samples. Could the authors apply a more thorough data analysis? I suggest to use two figures showing the spectra of the pristine and M-LRO samples overlapped, for the first at 4.55V and for the second one at 4.88V (between 525eV and 534eV). Also I want to add that the Figure 4c) is not sufficient to demonstrate e_g peak variations.

Reply: Thanks a lot. From Figures 4a and b, we can find that as the lithium ion is extracted, the change of crystal structure results to gradual weakening of the E_g peak. Until the lithium ion is completely extracted, the E_g peak becomes the weakest, as we emphasized in the text, the peak strength is very weak. In order to make a better comparison, the overlapped soft X-ray absorption spectrum curves of the two samples are shown in Supplementary Figure 12 in the two states of C4.55 V and C4.8 V. As the Figure shown, it is still obvious for the difference of E_g peak at the state of C4.55 V. Note that the difference of E_g peak is less obvious in fluorescence yield (FY) mode due to the structural change in C4.8 V. However, the peak still shows a strong signal in total electron yield (TEY) mode because of the difference in the detection mode. And the E_g peak of modified sample is stronger in TEY mode as shown by Supplementary Figure 11e and f. When discharged to 2.0V, the E_g peak of the modified sample can be recovered better than the pristine sample, which shows that the disturbance of the crystal structure is smaller on the modified sample when the lithium is completely removed. In summary, the analysis of both FY and TEY mode verified the stronger E_g peak for the modified sample, although the peak is relatively weaker at FY mode than that at the TEY mode.

Supplementary Figure 12. Soft XAS of both samples at charged 4.55 V and charged 4.8 V.

Supplementary Figure 11e and f. The soft XAS of O K-edge for both samples with TEY mode.

For the second question, Figure 4c does not focus on the change of E_g peak, instead of the integration of the soft X-ray spectra between 525 and 534 eV, which reflects the density of hole states just above the Fermi level (Nat. Chem. 2016, 8(7), 684-691 and J. Am. Chem. Soc. 2016, 138(35), 11211-11218). These oxygen holes are generated after the electrons escape, and their intensity is proportional to the oxygen redox chemistry. As described in the Figure 4c and Page 17, the intensity of $\text{Li}_{1.2}\text{Mn}_{0.6}\text{Ni}_{0.2}\text{O}_{2-\delta}$ sample is stronger than that of pristine sample, illustrating the activity of oxygen redox chemistry of the $\text{Li}_{1.2}\text{Mn}_{0.6}\text{Ni}_{0.2}\text{O}_{2-\delta}$ is stronger and it

can contribute more capacity. The corresponding modifications are added in highlighted part of Page 17 and Supplementary Figure 12 in the revised manuscript and supporting information.

Specific Comment No. 3: In Supplementary Figure 8, the PITT spectra of both samples during charging process are presented. This data analysis is superficial. The authors need to clearly discuss the result, especially, the higher value obtained for the M-LRO sample near 2.3V, and the Li diffusion coefficient variation for both samples which are different. The authors state: “electrochemical performance is in good agreement with the result of the PITT”. What is the reason for this in regard to this figure?

Reply: Thanks a lot for your suggestion. For the first point, we consider it to be an error point caused by the unstable voltage, but in order to show the complete result and it accords with the result that overall modified sample is better than pristine, so it is not deleted. Based on the reviewer’s comments, and a more rigorous explanation of the problem, we re-tested the lithium ion diffusion coefficient, as shown in new Supplementary Figure 8. The detailed discussion is as follows.

Supplementary Figure 8. PITT spectra of both samples during charging process.

The D_{Li^+} value of both samples decreased from the OCV to around 3.7 V during the first charge process, which may be caused by the narrowing Li^+ diffusion

channel with Li ions initial extraction consistent with the decreased c and a parameters according to previous paper (ACS Appl. Mater. Interfaces 2014, 6, 13271 – 13279). Between 3.8-4.5 V, the increase of the electrostatic repulsion between oxygen atoms with the further deintercalation of Li^+ , which is beneficial to the Li ions diffusion due to the expanded I_{LiO_2} layers. And because of the inherent larger I_{LiO_2} layers in the modified sample, the Li^+ diffusion coefficient of $\text{Li}_{1.2}\text{Mn}_{0.6}\text{Ni}_{0.2}\text{O}_{2-\delta}$ sample is better compared with that of the pristine. The decreased diffusion coefficient could be ascribed to the migration of additional nickel ions and structural rearrangement with the oxygen release at 4.6-4.8V, which also means the kinetics of the cationic redox chemistry before 4.5V is faster compared with anionic redox chemistry after voltage plateau. Specifically, the kinetics of oxygen redox is related with charge-transfer kinetics, and it stems from the distribution of charge-carrier $\text{Li}^+/\text{O}^{\text{n-}}$ is not like free molecule diffusion that can be simply described by Fick's law. Both the oxygen redox and Li^+ continuous migration will cause the local charge to be non-neutral. To maintain local charge neutrality, a charge redistribution/transfer should be accompanied when Li^+ and $\text{O}^{\text{n-}}$ are diffused or redistributed. Therefore, the chemical Li^+ diffusion coefficient determined here is also coupled with slow electron-transfer property that is sensitive to oxygen redox process, hence this is why there is a significantly decrease of coefficient after 4.5.

The larger lithium ions diffusion coefficient indicates that fast charging performance of the battery is better, which corresponds to the better rate capability, so we describe “The above electrochemical performance is in good agreement with the result of the potentiostatic intermittent titration technique (PITT)”. For more clarity, we change the description “electrochemical performance” to the specific “rate performance”. (Please see Highlights in Page 9 of the revised main manuscript and Page 9 of supporting information).

Figure 2a. Rate performance of both samples.

Specific Comment No. 4: On page 5, line 83/84, the authors state “It is worth noting that the LMRO with oxygen vacancies and/or spinel phase can improve the properties of cathode materials”. LMRO is not define in the main text. Could the authors specify this term? If it is the abbreviation of Li-rich Mn-based oxides, modify LMRO by LRMO in the whole manuscript.

Reply: Thanks a lot for the reminder of the reviewer. To make it easier to read, we have replaced all abbreviations of LRMO with $\text{Li}_{1.2}\text{Mn}_{0.6}\text{Ni}_{0.2}\text{O}_2$ in the main manuscript and replaced all abbreviations of M-LRO with $\text{Li}_{1.2}\text{Mn}_{0.6}\text{Ni}_{0.2}\text{O}_{2-\delta}$. The value of δ is 0.05 according to the result of neutron powder diffraction refinement. In addition, we have kept the corresponding abbreviations in the caption part, and where the abbreviation first appeared, the chemical formula was marked. The corresponding replacements have been highlighted throughout the revised manuscript.

Specific Comment No. 5: In the data analysis shown in Figure 4f) and h) (on pages 18-19), the authors state: “the peak intensity of M-LRO sample can revert back to the original state compared to the pristine sample” (lines 382-383), and “the d-d excitation peak of M-LRO sample can again revert back to the original intensity at OCV state, in contrast, the peak of pristine sample almost disappeared” (lines 394-396) I highly recommend to add “in the limit of the resolution of the ex situ Mn L-edge RIXS spectra”

because, the pristine spectrum obtained at D2.0V present a low signal to noise ratio.

Reply: Thanks a lot! We have added this recommend to the revised manuscript, the corresponding description is “When Li^+ are reinserted into the structure, the *d-d* excitation peak of $\text{Li}_{1.2}\text{Mn}_{0.6}\text{Ni}_{0.2}\text{O}_{2-\delta}$ sample can again revert back to the original intensity at OCV state, in contrast, the peak of pristine sample almost disappeared in the limit of the resolution of the ex situ Mn L-edge RIXS spectra,” (please see Highlight in Page 19 of the revised manuscript).

Specific Comment No. 6: On page 9, line 184/185, the authors state: “To reveal the mechanisms of the mitigated voltage decay and increased capacity. In-situ XRD is conducted to monitor the crystal structure change in real time”. I suggest to replace “capacity. In-situ” by “capacity, in-situ”.

Reply: Thanks a lot! We have revised the corresponding punctuation as suggested by the reviewer (Please see Highlight in Page 10 of the revised manuscript).

Specific Comment No. 7: On page 18, lines 371-373, the authors state: “To further illustrate the nature of enhanced reversible oxygen redox due to the oxygen vacancy and reduced Mn. The Mn L-edge RIXS of both samples are collected as Fig. 4f, h shown”. I suggest to replace “reduced Mn. The Mn” by “reduced Mn, the Mn”.

Reply: Thanks a lot! We have revised the corresponding punctuation as suggested by the reviewer (Please see Highlight in Page 18 of the revised manuscript).

Response to Reviewer #3:

General Comment: Non-hysteretic anionic redox reactions are important for designing high energy density cathode materials for lithium ion batteries. Authors reports on the reversible oxygen capacity in lithium rich LiMnO₂ cathode materials. Although the substitute of TM and O atom enhanced anion redox, those strategies have issues for practical usages, whereas the oxygen vacancy controlling d-d Coulomb interaction can improve the anion redox reversibility cost-effectively. This work attempted to demonstrate the fundamental understands of the oxygen redox. However, there are contradictions between this work and those published in the literature. It should be further clarified carefully.

Reply: We would like to thank reviewer for the comments, which are of great help to the improvement of the content of this article. Note that lots of reports of oxygen vacancies are beneficial to the electrochemical performance whether battery materials or catalysis materials (Nat. Commun. 2016, 7, 12108; Nat. Commun. 2018, 9, 1302; Nat. Mater. 2017, 16, 454-460), however, the mechanisms of oxygen vacancies are not been fundamentally explained. In this work, we have not only achieved excellent performance of a Li-rich layered oxide with oxygen vacancies, but we fundamentally understand the mechanism by advanced RIXS characterization technique. Next, we will revise and answer the questions according to the reviewer's comments one by one.

Specific Comment No. 1: The authors conclude that oxygen vacancy can enhance the anionic redox and mitigate the TM migration in this study. In Ref 17, the synthesize method and strategies are almost similar.

Reply: Thanks a lot! For this comment, we have a few points to explain. First, although these two papers are related to the oxygen vacancies, the synthesis method and mechanism study is different. In ref. 17 (J. Mater. Chem. A 2020, 8(16): 7733-7745) the material with oxygen vacancies is synthesized by CO₂ gas treatment in a tube furnace. For this method, we think it has a shortcoming, that is, vacancies can be generated well when the sample is in full contact with the gas. However, the

sample attached to the bottom of the quartz boat is not fully in contact with the gas, so the prepared sample contains uneven oxygen vacancies. In contrast, in this manuscript, we use the liquid-solid method to prepare sample with uniform oxygen vacancies, because after the pristine sample powder is dispersed in the hydrazine solution, it can be in good contact with the solution under stirring condition, so the prepared oxygen vacancies are more uniform.

Then, in the previous work, we analyzed the reasons for performance improvement from the following aspects: (1) oxygen vacancies can decrease the covalency of TM-O and mitigate oxygen gas release, (2) spinel with 3D Li⁺ diffusion channel can increase the rate capability and (3) MnO₆ octahedral distortion. Although these proposed mechanisms can well explain the reasons for the performance improvement and there are some innovations compared to the previous ones, we are still eager to explain the related phenomena more fundamentally. Fortunately, with the help of advanced characterization technology, especially RIXS technique, we have a fundamental understanding for oxygen vacancies from a more microscopic perspective, d-d Coulombic interaction. We believe that this mechanism will have a positive impact on other materials.

Specific Comment No. 2: The oxygen vacancy breaks the band of Li-O-Li, which reduces O₂ release. However, the oxygen defect can also effect on d-band sift of both Mn and Ni. In this study, authors discuss the electronic structures of Mn by experiments. The effects of d-band shift of Ni needs to be explained.

Reply: Thanks a lot for the comment. As we know, although in principle the X-ray absorption on the K-edge originates from the dipole-allowed transition from 1s to 4p empty orbital, the actual XANES spectrum not only reflects the electronic state of the 4p orbital near the Fermi level, but is also affected by the d band shift as mentioned by the reviewer, so it also includes various information about the electronic state and the local structure of absorbing atoms.

In this study, the initial cycle ex-situ XAS spectra of Ni K-edge of both samples

are shown in Supplementary Figure 10a and b. First, compared with NiO, the absorption edges of both samples are not completely overlapped with that of NiO due to the difference in the coordination structure, but the oxidation state of Ni shows a +2 valence. Then, with the gradual increase of voltage, the absorption edge of Ni K-edge also gradually moves toward the higher energy direction, corresponding to the oxidation of Ni²⁺ to Ni⁴⁺. Note that the absorption edges are similar at charged 4.3 V and at charged 4.8 V, indicating that the charge compensation of Ni mainly occurs before the 4.5 V voltage plateau in the layered Li-rich oxides. However, the difference between white line peaks of pristine sample Li_{1.2}Mn_{0.6}Ni_{0.2}O₂ when charged to 4.3 V and 4.8 V is more significant than that of the modified sample Li_{1.2}Mn_{0.6}Ni_{0.2}O_{2-δ}, which means that the local structure of Ni has changed during the oxidation process, and this may be caused by the migration of Ni from transition metal layer to the Li ions layer. In addition, when discharged to 2.0 V, the absorption edge of the pristine sample cannot coincide with the OCV state, while the edge of modified sample can perfectly overlap, which indicates that Ni irreversibly migrated from the transition metal layer to Li ion layer and was trapped in it during the charging-discharging process for the pristine sample. In comparison, Ni is confined in the form of NiO₆ octahedral distortion (only a weak white line peak change) due to the softening of the structure in the oxygen vacancy sample, resulting in the complete recovery of the structure at discharged 2.0 V.

Supplementary Figure 10c and d show the XAS of Ni K-edge at different cycles for Li_{1.2}Mn_{0.6}Ni_{0.2}O₂ and Li_{1.2}Mn_{0.6}Ni_{0.2}O_{2-δ}, respectively. From the first cycle of discharged to 2.0 V to the 10th cycle and then to the 50th cycle, the absorption edge and white line peak of the pristine Li_{1.2}Mn_{0.6}Ni_{0.2}O₂ gradually change, indicating the electronic state and local structure of Ni have changed in the pristine sample. While the modified sample hardly change, meaning Ni is stable in the transition metal layer in Li_{1.2}Mn_{0.6}Ni_{0.2}O_{2-δ}. In summary, oxygen vacancy and reduced Mn soften the crystal structure, which binds Ni to the transition metal layer, stabilizes the crystal structure, and improves the performance of the material. The detailed

explanation is added in the Page 11-12 of the revised Supporting Information.

Supplementary Figure 10. Ni K-edge spectra of both samples at different delithiation states and different cycles. (a and c) Pristine sample and (b and d) M-LRO sample. The spectrum of M-LRO sample can return to the OCV state when discharge to 2.0 V compared with the pristine sample. Especially, the spectra of M-LRO sample almost unchanged at first 50 cycles.

Specific Comment No. 3: The thermodynamic instability of oxygen defect would be expected. In this study, the oxygen vacancy reached about 5% in the sample. As is well-known, the defect of oxygen is related to crystal structure collapse. Authors can explain more details why the crystal changes in the modified sample is smaller than it in the pristine sample during lithiation?

Reply: Thanks a lot! We think an appropriate amount of oxygen vacancies will not affect structural integrity and stability of Li-rich layered oxides. In our research the oxygen vacancies is approximately 5% obtained from the Rietveld refinement of neutron diffraction, which content is similar with the previous report (Nat.

Commun. 2016, 7, 12108). Therefore, the stability and integrity of structure of modified sample $\text{Li}_{1.2}\text{Mn}_{0.6}\text{Ni}_{0.2}\text{O}_{2-\delta}$ are preserved. In addition, from the rate performance (Supplementary Figure 5d) of samples with different concentrations and different treatment time, it can be seen that too few oxygen vacancies (low concentration) cannot show the superiority of the defects, while too many oxygen vacancies (high concentration) will destroy its structural stability. However, we did not conduct neutron diffraction on both samples due to the limited time of large scientific devices. So we think that an oxygen vacancy content of approximately 5% is beneficial to the structural stability and integrity and the performance of materials.

Supplementary Figure 5d. The rate capacity of samples treated at different hydrazine concentrations and times.

In the process of lithiation and delithiation, the crystal structure of the material will shrink and expand, which can be seen from the change of peaks in in-situ XRD (Figure 3a and b). This continuous shrinkage and expansion will cause irreversible damage and transition metal migration in the crystal structure, and when the damage accumulates to a certain degree, the crystal structure will collapse. In this study, the oxygen vacancy and reduced Mn soften the crystal structure, which can be verified by the smaller change of peaks in in-situ XRD and can be verified by dislocation as shown in aberration-corrected scanning transmission electron microscopy (Supplementary Figure 2). This softened structure can withstand crystal changes and can limit the migration of transition metals in

the form of MO_6 octahedral reversible distortion, thereby stabilizing the crystal structure, which can also be proved by the Mn K-edge XAS (Figure 3c-f).

Figure 3a and b. In situ XRD patterns. (a) Pristine sample $\text{Li}_{1.2}\text{Mn}_{0.6}\text{Ni}_{0.2}\text{O}_2$, (b) modified sample $\text{Li}_{1.2}\text{Mn}_{0.6}\text{Ni}_{0.2}\text{O}_{2-\delta}$.

Supplementary Figure 2. The images of pristine (a and c) and M-LRO (b and d) sample from aberration-corrected scanning transmission electron microscopy.

Figure 3d-f. Mn K-edge XAS patterns. (c and e) $Li_{1.2}Mn_{0.6}Ni_{0.2}O_2$, (d and f) $Li_{1.2}Mn_{0.6}Ni_{0.2}O_{2-\delta}$.

In addition, the oxygen vacancy can mitigate oxygen release during charging and discharging, which can be verified by the DEMS (Supplementary Figure 16). On the one hand, less oxygen release reduces the collapse of crystal structure due to the lack of coordinated oxygen. On the other hand, more oxygen can react more completely with electrolyte, the by-products (HF and oxygen radicals etc.) produced in this process will corrode and attack the material crystal, so the damage to the crystal structure is more serious. In summary, the soften structure and reduced oxygen release decrease the crystal change in the modified sample.

Supplementary Figure 16. Oxygen release of pristine (a) and M-LRO ($\text{Li}_{1.2}\text{Mn}_{0.6}\text{Ni}_{0.2}\text{O}_{2-\delta}$) (b) samples measured by operando DEMS.

Specific Comment No. 4: If the oxygen vacancies are generated more ($>5\%$), does the electronic state of anion have similar energy state of TM?

Reply: Thanks a lot! We think it has the similar electronic energy states. First, when the lattice oxygen does not form O_2 , whether there is oxygen vacancy or not, the lattice oxygen forms a covalent bond with TM. Therefore, the 2p-electron of the lattice oxygen and the 3d-electron of TM are in a hybrid state, so their electronic state is similar. Then, when the similar oxygen molecule O_2 is formed, the hybridization of O and TM will be greatly reduced, and at the same time, there will be obvious isolated O_2 energy level characteristic peaks on the DOS diagram. In the calculations in this article, the concentration of oxygen vacancies is $3/54 * 100\% = 5.6\%$. It can be seen from the optimized structure (Supplementary Figure 20) and DOS figure (Supplementary Figure 23) that there is no isolated characteristic peak of O_2 in the structure. Therefore, the remaining lattice oxygen is still hybridized with TM to form bonds and have similar electronic states.

Supplementary Figure 20. The DFT calculation models. (a) pristine sample and (b) M-LRO sample.

Supplementary Figure 23. The DOS patterns of both samples at different charge-discharge states.

REVIEWERS' COMMENTS

Reviewer #1 (Remarks to the Author):

The authors reply to all my comments/concerns.
To my opinion, the manuscript is now ready for publication.

Reviewer #2 (Remarks to the Author):

The clarifications regarding the data analysis are sufficient for readers to understand. The manuscript can be published in its present form.

Reviewer #3 (Remarks to the Author):

The manuscript has been revised accordingly. There is no more question.